# CREAM: Consistency Regularized Self-Rewarding Language Models

**Zhaoyang Wang**[1]  **Weilei He**[2]  **Zhiyuan Liang**[3]  **Xuchao Zhang**[4]
**Chetan Bansal**[4]  **Ying Wei**[2]  **Weitong Zhang**[1]  **Huaxiu Yao**[1]
[1]University of North Carolina at Chapel Hill  [2]Nanyang Technological University
[3]National University of Singapore  [4]Microsoft Research
{zhaoyang,huaxiu}@cs.unc.edu  weitongz@unc.edu

## ABSTRACT

Recent self-rewarding large language models (LLM) have successfully applied LLM-as-a-Judge to iteratively improve the alignment performance without the need of human annotations for preference data. These methods commonly utilize the same LLM to act as both the policy model (which generates responses) and the reward model (which scores and ranks those responses). The ranked responses are then used as preference pairs to train the LLM via direct alignment technologies (e.g. DPO). However, it is noteworthy that throughout this process, there is no guarantee of accuracy in the rewarding and ranking, which is critical for ensuring accurate rewards and high-quality preference data. Empirical results from relatively small LLMs (e.g., 7B parameters) also indicate that improvements from self-rewarding may diminish after several iterations in certain situations, which we hypothesize is due to accumulated bias in the reward system. This bias can lead to unreliable preference data for training the LLM. To address this issue, we first formulate and analyze the generalized iterative preference fine-tuning framework for self-rewarding language model. We then introduce the regularization to this generalized framework to mitigate the overconfident preference labeling in the self-rewarding process. Based on this theoretical insight, we propose a **C**onsistency **R**egularized s**E**lf-rewarding l**A**nguage **M**odel (CREAM) that leverages the consistency of rewards across different iterations to regularize the self-rewarding training, helping the model to learn from more reliable preference data. With this explicit regularization, our empirical results demonstrate the superiority of CREAM in improving both reward consistency and alignment performance. The code is publicly available at https://github.com/Raibows/CREAM.

## 1 INTRODUCTION

Large language models (LLMs) have shown impressive capabilities across various tasks, including natural language understanding and generation (Radford et al., 2019). At the same time, LLMs also face alignment challenges such as generating hallucinations and harmful outputs (Ji et al., 2023). To address these issues, a series of research works has explored preference learning methods such as Reinforcement Learning from Human Feedback (RLHF) (Christiano et al., 2017) and direct alignment techniques such as Direct Preference Optimization (DPO) (Rafailov et al., 2023) to align the LLMs with human values and preferences. These alignment methods often require a large number of preference pairs which are indispensable in both RLHF and direct alignment training. However, collecting human-annotated preference pairs is time-consuming and labor-intensive, which seriously limits the scalability and efficiency of these alignment methods.

Recent advancements in self-rewarding language models (SRLMs) (Yuan et al., 2024) have attracted increasing attention in the field of LLM alignment, which can efficiently synthesize preference data for iterative preference training. In this method, a single LLM is required to act as two roles, the policy model and the reward model. Given unlabeled prompt data, the LLM first acts as the policy model generating several response candidates. Then, the same LLM acts as the reward model, scoring and ranking these responses. These ranked responses are used as preference pairs to train the LLM with DPO, significantly reducing the reliance on human-annotated data. The above steps can

be iteratively repeated to further enhance the performance. However, SRLMs still face challenges in generating reliable and accurate rewards for annotating the preference pairs, which is critical for ensuring both the quality of preference data and the alignment performance of LLMs.

To address these challenges, we first formulate a generalized iterative preference fine-tuning framework to analyze the self-rewarding training, where this framework can also be adapted to other iterative preference tuning methods. Through this theoretical framework, we find that the rewarding bias issue in SRLMs arises from the overconfident preference labeling, which enforces the model to distinguish between responses of similar quality. For example, both responses in Figure 1 have high quality judgments from the human. The SRLM enforces the reward model to make a preference judgment, resulting in noisy and unreliable preference labeling. This can lead to negative impacts on preference tuning the model. Additionally, the iterative training manner can also accumulate the rewarding bias, further diminishing the benefits of self-improvement.

From the insights of theoretical analysis, we propose **C**onsistency **R**egularized s**E**lf-rewarding l**A**nguage **M**odel (CREAM) to mitigate the rewarding bias issue in SRLMs, particularly for broadly accessible 7B-size LLMs. The core idea behind CREAM is that we should not force the model to be overly confident in distinguishing between responses of similar quality. *But how to tell the preference labeling is reliable or not?* Out of the self-rewarding scenario, we may employ a pool of external reward models to assist in ranking preferences. When two responses are of similar quality, these external models often produce inconsistent rankings.

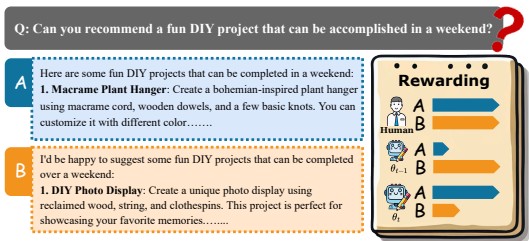

Figure 1: An example of both two responses are of high quality, which is hard for human to distinguish the preference. While the same model from different iterations have inconsistent rewarding.

This inconsistency serves as a signal to indicate the level of confidence in the preference labeling. In self-rewarding scenarios, however, integrating such external reward models is not feasible. Fortunately, due to the iterative nature of self-rewarding training, we can use the reward model from the previous iteration to rank preferences and then compare these rankings with those produced by the current model. This comparison provides an estimate of such consistency rate. With this consistency rate, we can regularize the preference training to prevent the model from learning unreliable preference data, thereby mitigating the rewarding bias issue in SRLMs.

In summary, we first formulate a generalized iterative preference fine-tuning framework to analyze the rewarding bias issue in SRLMs. From the insights of theoretical analysis, we propose CREAM as the primary contribution of this paper. CREAM leverages the consistency of rewarding across different iterations for regularized preference training, which can effectively mitigate the rewarding bias issue in SRLMs. Empirical results on a series of natural language benchmarks validate the effectiveness of CREAM in mitigating the rewarding bias issue and enhancing the alignment performance of LLMs.

**Notations.** Vectors are denoted by lowercase boldface letters, such as $\mathbf{x}$, and matrices by uppercase boldface letters, such as $\mathbf{A}$. For any positive integer $k$, the set $1, 2, \ldots, k$ is denoted by $[k]$. Other general sets are denoted by calligraphic uppercase letters, such as $\mathcal{D}$, with the cardinality of the set represented as $|\mathcal{D}|$. Without ambiguity, we denote $\pi_{\boldsymbol{\theta}}$ as the language model parameterized by $\boldsymbol{\theta}$, $\mathbf{x}$ as the input prompt, and $\mathbf{y}$ as the output response from the language model. All other notations are defined prior to their first usage. We denote $\mathbb{1}[\cdot]$ as the indicator function.

## 2  RELATED WORKS

**LLM Alignment.** Alignment lies at the core of LLM research and applications, aiming to ensure that LLMs adhere to human values and preferences. RLHF establishes the foundational alignment training paradigm (Leike et al., 2018; Ziegler et al., 2019; Ouyang et al., 2022a), where it leverages human preference feedback to train a reward model, and then uses this reward model to guide the LLM via reinforcement learning algorithms (Schulman et al., 2017). Recent efforts have focuses on developing direct alignment methods (Rafailov et al., 2023; Dong et al., 2023; Azar et al., 2023; Ethayarajh et al., 2024; Meng et al., 2024; Hong et al., 2024), in order to reduce the costs and

complexity of RLHF and make it more efficient and accessible. DPO (Rafailov et al., 2023), as a representative direct alignment method, optimizes the LLM with annotated preference pairs, eliminating the need for training an additional reward model. However, most RLHF and direct alignment methods heavily rely on human-annotated preference data, where the data collection commonly involves human distinguishing the "good" responses from the "bad" ones, which is time-consuming and labor-intensive (Ouyang et al., 2022a; Bai et al., 2022). Thus, synthesizing preference data with minimal human effort has become a valuable research direction.

**Self-Rewarding Language Model.** SRLM (Yuan et al., 2024) has emerged as a promising approach to address the challenge of preference data synthesis in a self-improvement manner. This method leverages the LLM itself to act as both the policy model and the reward model. The policy model can generate response candidates for unlabeled prompts, while the reward model uses LLM-as-A-Judge (Zheng et al., 2023; Bai et al., 2023; Dubois et al., 2024) prompting to reward and rank these responses based on their quality. The ranked responses are then used as preference pairs to train the LLM via DPO (Rafailov et al., 2023). And this process can be iteratively repeated to improve the alignment performance without human intervention. However, having the same LLM serve as both the policy and the reward model, without any regularization, presents challenges in guaranteeing accurate rewards. This can lead to bias accumulation and noisy preference data, which ultimately harms the training. Other similar self-improvement methods (Huang et al., 2022; Zelikman et al., 2022; Chen et al., 2024; Guo et al., 2024b; Zhou et al., 2024) often either use the ground truth response to avoid annotation bias, or introduce an additional reward model to reduce the noise in annotations. In contrast, our work neither requires labeled data nor relies on external LLMs. Instead, we propose to use the consistency of rewarding to mitigate the rewarding bias in SRLMs.

**Reward Hacking.** In both RLHF and SRLM scenarios, the reward model plays a crucial role in training LLMs (Ouyang et al., 2022b; Anwar et al., 2024; Yuan et al., 2024; Fisch et al., 2024). For RLHF, reward hacking is a phenomenon where models exploit flaws or biases in reward models to maximize scores without aligning with the intended goals (Anwar et al., 2024). To mitigate this issue, various ensemble rewarding methods (Coste et al., 2023; Eisenstein et al., 2023; Ramé et al., 2024; Zhang et al., 2024) such as ensemble-based conservative optimization (Coste et al., 2023) and averaging rewards in the weight space (Ramé et al., 2024) have been proposed to improve reliability and robustness. However, these works mainly focused on estimating the rewards, while CREAM in the self-rewarding scenario only uses rewards for comparing the responses to annotate preference pairs instead of maximizing the rewards. Besides, CREAM uses regularization instead of conservative value estimation to mitigate the rewarding bias issue. Though applying ensemble rewarding methods such as robust preference optimization (Fisch et al., 2024) which proposes reward distillation and pessimistic ensemble is feasible, CREAM does not rely on ensemble rewarding which reduces the computational cost in SRLMs. Additionally, CREAM leverages the iterative nature of SRLM to estimate the consistency between iterations to regularize the preference pair, which is more suitable for iterative preference fine-tuning. Further discussion is in Appendix C.4.

## 3 METHODOLOGY

In this section, we first formulate the generalized iterative preference fine-tuning framework for self-rewarding, RL with AI feedback, and other iterative preference tuning methods. Next, we introduce the motivation behind the proposed consistency regularized self-rewarding method. Finally, we present the practical implementation algorithm of CREAM in detail.

### 3.1 GENERALIZED ITERATIVE PREFERENCE FINE-TUNING FRAMEWORK

We assume that we can access to the dataset with response $\mathcal{D}_\mathrm{S}$ and the prompt dataset without response $\mathcal{D}_\mathrm{U}$. The objective is to iteratively minimize the following loss with respect to the neural network parameter $\boldsymbol{\theta}$ and a label function $z$ as

$$\mathcal{L}(\boldsymbol{\theta}, z) = \mathcal{L}_{\mathrm{SFT}}(\boldsymbol{\theta}; \mathcal{D}_\mathrm{S}) + \mathbb{E}_{\mathbf{x} \sim \mathcal{D}_\mathrm{U}; \mathbf{y}, \mathbf{y}' \sim \pi_{\boldsymbol{\theta}_t}(\cdot|\mathbf{x})}[\mathcal{L}_{\mathrm{DPO}}(\boldsymbol{\theta}; \mathbf{y}, \mathbf{y}', \mathbf{x}, z)]. \tag{3.1}$$

where the first term $\mathcal{L}_{\mathrm{SFT}}(\boldsymbol{\theta}; \mathcal{D}_\mathrm{S})$ aligns the model $\pi_{\boldsymbol{\theta}}$ to the SFT data. We note here that any potential SFT methods (Ouyang et al., 2022b; Yuan et al., 2023; Dong et al., 2023; Chen et al., 2024), or the methods without SFT data ($\mathcal{L}_{\mathrm{SFT}} = 0$) can be adapted in this framework. The second term $\mathbb{E}[\mathcal{L}_{\mathrm{DPO}}]$ corresponds to learning from the preference data pair $\{\mathbf{y}, \mathbf{y}'\}$ generated by the current model $\boldsymbol{\theta}_t$. The labeling function $z(\mathbf{y}, \mathbf{y}', \mathbf{x}) \in \{0, 1\}$ provides the preference judgment between $\mathbf{y}$ and $\mathbf{y}'$ for

the DPO loss, where $z(\mathbf{y}, \mathbf{y}', \mathbf{x}) = 1$ means $\mathbf{y} \succ \mathbf{y}'$ and $z(\mathbf{y}, \mathbf{y}', \mathbf{x}) = 0$ means $\mathbf{y} \prec \mathbf{y}'$. The DPO loss $\mathcal{L}_{\text{DPO}}$ is defined as follows:

$$\mathcal{L}_{\text{DPO}}(\boldsymbol{\theta}; \mathbf{y}, \mathbf{y}', \mathbf{x}, z) = -z(\mathbf{y}, \mathbf{y}', \mathbf{x}) \log \sigma \left( \log \left( \frac{\pi_{\boldsymbol{\theta}}(\mathbf{y}|\mathbf{x})}{\pi_{\text{ref}}(\mathbf{y}|\mathbf{x})} \right) - \log \left( \frac{\pi_{\boldsymbol{\theta}}(\mathbf{y}'|\mathbf{x})}{\pi_{\text{ref}}(\mathbf{y}'|\mathbf{x})} \right) \right)$$
$$- (1 - z(\mathbf{y}, \mathbf{y}', \mathbf{x})) \log \sigma \left( \log \left( \frac{\pi_{\boldsymbol{\theta}}(\mathbf{y}'|\mathbf{x})}{\pi_{\text{ref}}(\mathbf{y}'|\mathbf{x})} \right) - \log \left( \frac{\pi_{\boldsymbol{\theta}}(\mathbf{y}|\mathbf{x})}{\pi_{\text{ref}}(\mathbf{y}|\mathbf{x})} \right) \right), \quad (3.2)$$

where $\pi_{\text{ref}}$ is the reference model for KL divergence regularization, and $\sigma(\cdot)$ is the sigmoid function. The proposed loss $\mathcal{L}(\boldsymbol{\theta}, z)$ in Eq. (3.1) represents all iterative preference fine-tuning algorithms. For the reinforcement learning (RL) with human feedback (Ouyang et al., 2022b), $z$ is the human preference comparing $\mathbf{y}$ and $\mathbf{y}'$. For the RL with AI feedback, $z$ is the oracle reward model like GPT-4 (Achiam et al., 2023). For the self-rewarding language model (Chen et al., 2024), $z$ is given by comparing the reward score generated from the language model itself, often with LLM-as-a-Judge prompting. However, as aforementioned, we note that such prompt rewarding method may only be feasible for larger and advanced LLMs such as Llama-70B (Touvron et al., 2023). For smaller models such as Llama-7B that do not have complex instruction following and reasoning abilities, we instead propose to leverage the intrinsic reward model (Rafailov et al., 2023)

$$r_{\boldsymbol{\theta}}(\mathbf{x}, \mathbf{y}) \propto [\log \pi_{\boldsymbol{\theta}}(\mathbf{y}|\mathbf{x}) - \log \pi_{\text{ref}}(\mathbf{y}|\mathbf{x})]$$

to reward and rank the responses for annotating preference pairs. Therefore, the choice of preference labeling function $z$ is closely connected with the language model parameter $\boldsymbol{\theta}$. Then, we introduce the following two-step optimization algorithm to solve Eq. (3.1).

**Step 1.** (Preference-labeling step) Keep $\boldsymbol{\theta} = \boldsymbol{\theta}_t$ fixed, select function $z$ to minimize $\mathcal{L}_{\text{DPO}}$. In particular, letting $\boldsymbol{\theta} = \boldsymbol{\theta}_t$ in Eq. (3.2), solution for $z(\mathbf{y}, \mathbf{y}', \mathbf{x}) = \arg\min_z \mathcal{L}_{\text{DPO}}(\boldsymbol{\theta}_t; \mathbf{y}, \mathbf{y}', \mathbf{x}, z)$ is

$$z_{t+1}(\mathbf{y}, \mathbf{y}', \mathbf{x}) = \mathbb{1}\left[\log \pi_{\boldsymbol{\theta}_t}(\mathbf{y}|\mathbf{x}) - \log \pi_{\text{ref}}(\mathbf{y}|\mathbf{x}) \geq \log \pi_{\boldsymbol{\theta}_t}(\mathbf{y}'|\mathbf{x}) - \log \pi_{\text{ref}}(\mathbf{y}'|\mathbf{x})\right]. \quad (3.3)$$

**Step 2.** (Learning step) Keep $z$ as of Eq. (3.3), minimize loss function $\mathcal{L}(\boldsymbol{\theta}, z_{t+1})$ with respect to $\boldsymbol{\theta}$ and get $\boldsymbol{\theta}_{t+1} = \arg\min_{\boldsymbol{\theta}} \mathcal{L}(\boldsymbol{\theta}, z_{t+1})$.

Different from existing methods, the proposed two-step optimization method directly uses the intrinsic reward model to generate the preference data. This approach is particularly feasible for small LLMs, which lack the capacity to effectively use LLM-as-a-Judge prompts (Zheng et al., 2023) for rewarding and ranking. We note that the proposed two-step method is similar to the Expectation-Maximization algorithm and self-training paradigm (Zou et al., 2019). This similarity is supported by the following theorem, which suggests the convergence of the proposed two-step algorithm.

**Theorem 3.1.** Suppose the optimization $\boldsymbol{\theta}_{t+1} = \arg\min_{\boldsymbol{\theta}} \mathcal{L}(\boldsymbol{\theta}, z_{t+1})$ is solvable and the SFT loss $\mathcal{L}_{\text{SFT}}(\boldsymbol{\theta}; \mathcal{D}_{\text{S}}) \geq 0$ for all $\boldsymbol{\theta}$ and $\mathcal{D}_{\text{S}}$, the proposed two-step optimization method converges.

## 3.2 CONSISTENCY REGULARIZED SELF-REWARDING

The generalized framework presented in Eq. (3.1) assumes the human feedback or GPT-4 are all reliable so that the preference labeling function $z$ is trustworthy. However, for SRLMs, the accuracy of preference labeling is not always guaranteed. Therefore, treating all selected preference labels as "ground truth" by encoding them as hard labels can lead to overconfident mistakes, potentially propagating biases and inaccuracies from the LLMs. Taking Figure 1 as an example, both the two responses $\mathbf{y}$ and $\mathbf{y}'$ are judged by humans to be of high quality. *Forcing the model to be overly confident in distinguishing between these two responses $\{\mathbf{y}, \mathbf{y}'\}$ of similar quality can negatively impact the performance of SRLMs during training.*

This rewarding bias issue motivates us to mitigate such ambiguity by introducing a consistency-regularized self-rewarding language model, CREAM. Specifically, for a pair of responses with very similar quality, their oracle reward scores should ideally be very close to each other. Particularly, when multiple reward models are available, it is likely that some models will rank one response as superior, while others may rank the opposite response as better, resulting in high ranking inconsistency (i.e., low ranking consistency) among these models. Based on this, CREAM aims to prevent the model from learning from preference pairs with low consistency. Instead, it focuses

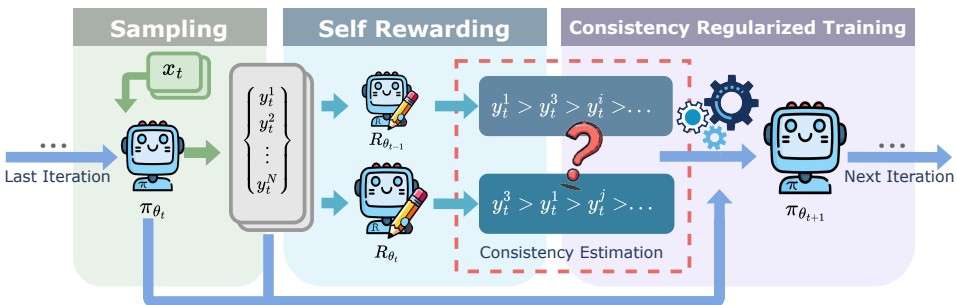

Figure 2: The flow of CREAM. In the response sampling stage, the policy model $\pi_{\boldsymbol{\theta}_t}$ generates $N$ responses. After that, CREAM uses the reward model $R_{\boldsymbol{\theta}_{t-1}}$ from the previous iteration to reward and rank these responses. Then, the rankings are compared with those generated by current reward model $R_{\boldsymbol{\theta}_t}$ to estimate the consistency rate. Finally, the policy model $\pi_{\boldsymbol{\theta}_t}$ is fine-tuned with consistency regularized preference training objective, resulting in the model $\pi_{\boldsymbol{\theta}_{t+1}}$ for next iteration.

solely on preference pairs with high consistency across different reward models, thereby mitigating the rewarding bias issue and stabilize the learning process to some extent. From the theoretical perceptive, we can introduce a regularization term to Eq. (3.1) as

$$\mathcal{L}(\boldsymbol{\theta}, z) = \mathcal{L}_{\text{SFT}}(\boldsymbol{\theta}; \mathcal{D}_{\text{S}}) + \mathbb{E}_{\mathbf{x} \sim \mathcal{D}_{\text{U}}; \mathbf{y}, \mathbf{y}' \sim \pi_{\boldsymbol{\theta}_t}(\cdot|\mathbf{x})}[\mathcal{L}_{\text{DPO}}(\boldsymbol{\theta}; \mathbf{y}, \mathbf{y}', \mathbf{x}, z) + \lambda \mathcal{L}_{\text{Reg}}(\boldsymbol{\theta}; \mathbf{y}, \mathbf{y}', \mathbf{x})], \quad (3.4)$$

where the regularization term $\mathcal{L}_{\text{Reg}}(\boldsymbol{\theta}; \mathbf{y}, \mathbf{y}', \mathbf{x})$ prevents the model $\pi_{\boldsymbol{\theta}}$ from overconfidence in distinguishing the preference of $\{\mathbf{y}, \mathbf{y}'\}$ of similar quality, which is quantified in the following lemma.

**Lemma 3.2.** Let the random variable $z = z(\mathbf{y}, \mathbf{y}', \mathbf{x})$ be defined as $z(\mathbf{y}, \mathbf{y}', \mathbf{x}) = \mathbb{1}[\mathbf{y} \succ \mathbf{y}'|\mathbf{x}]$. The Bradley-Terry model (Bradley & Terry, 1952) for the probability of $z$ under parameter $\boldsymbol{\theta}$ is given by

$$P_{\boldsymbol{\theta}}(z) = P_{\boldsymbol{\theta}}(\mathbb{1}[\mathbf{y} \succ \mathbf{y}'|\mathbf{x}]) = \sigma \left( \log(\pi_{\boldsymbol{\theta}}(\mathbf{y}|\mathbf{x})/\pi_{\text{ref}}(\mathbf{y}|\mathbf{x})) - \log(\pi_{\boldsymbol{\theta}}(\mathbf{y}'|\mathbf{x})/\pi_{\text{ref}}(\mathbf{y}'|\mathbf{x})) \right),$$

Letting the regularization $\mathcal{L}_{\text{Reg}}$ be defined by

$$\begin{aligned}
\mathcal{L}_{\text{Reg}}(\boldsymbol{\theta}; \mathbf{y}, \mathbf{y}', \mathbf{x}) = & -\log \sigma \left( \log(\pi_{\boldsymbol{\theta}}(\mathbf{y}|\mathbf{x})/\pi_{\text{ref}}(\mathbf{y}|\mathbf{x})) - \log(\pi_{\boldsymbol{\theta}}(\mathbf{y}'|\mathbf{x})/\pi_{\text{ref}}(\mathbf{y}'|\mathbf{x})) \right) \\
& -\log \sigma \left( (\log \pi_{\boldsymbol{\theta}}(\mathbf{y}'|\mathbf{x})/\pi_{\text{ref}}(\mathbf{y}'|\mathbf{x})) - (\log \pi_{\boldsymbol{\theta}}(\mathbf{y}|\mathbf{x})/\pi_{\text{ref}}(\mathbf{y}|\mathbf{x})) \right).
\end{aligned} \quad (3.5)$$

Then the expected regularized loss under the model $\boldsymbol{\theta}_t$ is given by:

$$\mathbb{E}_{\mathbf{y}, \mathbf{y}' \sim \pi_{\boldsymbol{\theta}_t}(\cdot|\mathbf{x})} \mathcal{L}_{\text{reg}}(\boldsymbol{\theta}; \mathbf{y}, \mathbf{y}', \mathbf{x}) = 2 \, \mathbb{KL}(u(\cdot) \| P_{\boldsymbol{\theta}}(\cdot)), \quad (3.6)$$

where $u(z)$ is the uniform binary distribution, i.e., $u(z = 0) = u(z = 1) = 0.5$.

As Lemma 3.2 suggests, the $\mathcal{L}_{\text{Reg}}$ will regularize the preference between $\{\mathbf{y}, \mathbf{y}'\}$ that has similar quality to a uniform distribution. Then the following theorem suggests that using $\mathcal{L}_{\text{DPO}} + \lambda \mathcal{L}_{\text{Reg}}$ corresponds to the soft-labeled DPO which we implemented in CREAM.

**Theorem 3.3.** For all $\mathbf{y}, \mathbf{y}', \mathbf{x}, z$, minimizing

$$\mathcal{L}(\boldsymbol{\theta}, z) = \mathcal{L}_{\text{SFT}}(\boldsymbol{\theta}; \mathcal{D}_{\text{SFT}}) + \mathbb{E}_{\mathbf{x} \sim \mathcal{D}_{\text{U}}; \mathbf{y}, \mathbf{y}' \sim \pi_{\boldsymbol{\theta}_t}(\cdot|\mathbf{x})} \left[ \mathcal{L}_{\text{DPO}}(\boldsymbol{\theta}; \mathbf{y}, \mathbf{y}', \mathbf{x}, z) + \lambda \mathcal{L}_{\text{Reg}}(\boldsymbol{\theta}; \mathbf{y}, \mathbf{y}', \mathbf{x}) \right]$$

is equivalent with minimizing

$$\begin{aligned}
\mathcal{L}(\boldsymbol{\theta}, z) = & \frac{1}{1 + 2\lambda} \mathcal{L}_{\text{SFT}}(\boldsymbol{\theta}; \mathcal{D}_{\text{S}}) \\
& + \mathbb{E}_{\mathbf{x} \sim \mathcal{D}_U; \mathbf{y}, \mathbf{y}' \sim \pi_{\boldsymbol{\theta}_t}(\cdot|\mathbf{x})}[\mathcal{C}_\lambda \mathcal{L}_{\text{DPO}}(\boldsymbol{\theta}; \mathbf{y}, \mathbf{y}', \mathbf{x}, z) + (1 - \mathcal{C}_\lambda)\mathcal{L}_{\text{DPO}}(\boldsymbol{\theta}; \mathbf{y}, \mathbf{y}', \mathbf{x}, 1 - z)],
\end{aligned} \quad (3.7)$$

where the $1 - z$ reverses the preference order of $z(\mathbf{y}, \mathbf{y}', \mathbf{x})$ and $\mathcal{C}_\lambda = (1 + \lambda)/(1 + 2\lambda)$. We choose to reverse the preference order if there is evidence that the annotated preference data is reversed. And the final form can also be viewed as the label smoothing. Details are in Appendix C.2.

Theorem 3.3 suggests that instead of calculating the regularization term $\mathcal{L}_{\text{Reg}}$, we can use the soft-labeled DPO to train Eq. (3.7). In particular, when $\lambda = 0$, $\mathcal{C}_\lambda = 0$ and Eq. (3.7) degenerates to Eq. (3.1). This represents the case where the preference label $z$ is trustworthy from human or

---

**Algorithm 1** Consistency-Regularized Self-Rewarding Language Model

---

**Input:** seed SFT dataset $\mathcal{D}_S$; unlabeled prompt dataset $\mathcal{D}_U$; initial model parameter $\boldsymbol{\theta}_0$;
**Input:** number of iterations $T$; learning rate $\eta$
1: /* SFT training */
2: Obtain $\boldsymbol{\theta}_1$ by taking the gradient steps over loss $L_1(\boldsymbol{\theta}) = \sum_{(\mathbf{x},\mathbf{y}) \in \mathcal{D}_S} \log \pi_{\boldsymbol{\theta}}(\mathbf{y}|\mathbf{x})$ from $\boldsymbol{\theta}_0$
3: /* Iterative Preference Training training */
4: **for** $t = 1$ to $T$ **do**
5:     Sample $\{\mathbf{y}_{ij}\}_{i=1}^N \sim \pi_{\boldsymbol{\theta}_t}(\cdot|\mathbf{x}_j)$ for all $\mathbf{x}_j \in \mathcal{D}_U$   // Response Sampling
6:     Compute reward $r_{ij} = \log \pi_{\boldsymbol{\theta}_t}(\mathbf{y}_{ij}|\mathbf{x}_i) - \log \pi_{\boldsymbol{\theta}_0}(\mathbf{y}_{ij}|\mathbf{x}_i)$ for all $i \in [N], j \in [|\mathcal{D}_U|]$
7:     Obtain rank $J_{ij}$ for all $y_{ij}$ using reward $r_{ij}$   // Rank on model $\boldsymbol{\theta}_t$
8:     Compute reward $r'_{ij} = \log \pi_{\boldsymbol{\theta}_{t-1}}(\mathbf{y}_{ij}|\mathbf{x}_i) - \log \pi_{\boldsymbol{\theta}_0}(\mathbf{y}_{ij}|\mathbf{x}_i)$ for all $i \in [N], j \in [|\mathcal{D}_U|]$
9:     Obtain rank $K_{ij}$ for all $y_{ij}$ using reward $r'_{ij}$   // Rank on model $\boldsymbol{\theta}_{t-1}$
10:     Compute $\tau_j = \tau(\{J_{ij}\}_i, \{K_{ij}\}_i)$ according to Eq. (3.10) for all $j \in [|\mathcal{D}_U|]$
11:     Compute consistency rate $\mathcal{C} = |\mathcal{D}_U|^{-1} \sum_j (\tau_j + 1)/2$ // Adaptive consistency regularization
12:     Compose preference dataset $\mathcal{D}_{\text{DPO}}$ using pairs $\{\mathbf{x}_j, \mathbf{y}_j^+, \mathbf{y}_j^-\}_j$ according to Eq. (3.11)
13:     Compose preference dataset $\mathcal{D}_{\text{RDPO}}$ using pairs $\{\mathbf{x}_j, \mathbf{y}_j^-, \mathbf{y}_j^+\}_j$ according to Eq. (3.12)
14:     Update $\boldsymbol{\theta}_{t+1}$ by minimizing loss $\mathcal{L}(\boldsymbol{\theta}) = \mathcal{C}\mathcal{L}_{\text{DPO}}(\pi_{\boldsymbol{\theta}_t}, \mathcal{D}_{\text{DPO}}) + (1 - \mathcal{C})\mathcal{L}_{\text{DPO}}(\pi_{\boldsymbol{\theta}_t}, \mathcal{D}_{\text{RDPO}})$
15: **end for**
**Output:** aligned policy model $\pi_{\boldsymbol{\theta}_T}$

---

some oracle reward models (e.g., GPT-4). In other words, $\lambda$ represents the *confidence* of the label function $z$. Specially, since in our two-step optimization paradigm, the label function $z$ is directly derived from the previous model $\pi_{\boldsymbol{\theta}_t}$, we can measure the performance of $\pi_{\boldsymbol{\theta}_t}$ using the consistency between model $\boldsymbol{\theta}_t$ and the baseline model (e.g., external reward model) $\boldsymbol{\theta}'_t$, defined by

$$\lambda(\mathbf{x}) = 2\mathbb{E}_{\mathbf{y},\mathbf{y}' \sim \pi_{\boldsymbol{\theta}_t}(\cdot|\mathbf{x})} \mathbb{1}[\mathbf{y} \succ \mathbf{y}'|\mathbf{x}, \boldsymbol{\theta}_t] \mathbb{1}[\mathbf{y} \succ \mathbf{y}'|\mathbf{x}, \boldsymbol{\theta}'_t], \tag{3.8}$$

and when $\lambda \to 0$, $\mathcal{C}_\lambda \approx 1 - \lambda$ representing the consistency of model $\boldsymbol{\theta}_t$ and $\boldsymbol{\theta}'_t$. $\mathbb{1}[\mathbf{y} \succ \mathbf{y}'|\mathbf{x}, \boldsymbol{\theta}_t]$ means the response $\mathbf{y}$ is better than $\mathbf{y}'$ given the prompt $\mathbf{x}$ and language model parameter $\boldsymbol{\theta}_t$, i.e.,

$$\mathbb{1}[\log(\pi_{\boldsymbol{\theta}_t}(\mathbf{y}|\mathbf{x})/\pi_{\text{ref}}(\mathbf{y}|\mathbf{x})) - \log(\pi_{\boldsymbol{\theta}_t}(\mathbf{y}'|\mathbf{x})/\pi_{\text{ref}}(\mathbf{y}'|\mathbf{x}))],$$

and similar definition applies to $\mathbb{1}[\mathbf{y} \succ \mathbf{y}'|\mathbf{x}, \boldsymbol{\theta}'_t]$.

## 3.3 Proposed Algorithm

Equipped with the above two-stage optimization and the consistency-regularized self-rewarding, we are ready to present the implementation of CREAM in Algorithm 1. The whole framework of CREAM is also illustrated in Figure 2. The algorithm starts from the SFT training to obtain the first model parameter $\boldsymbol{\theta}_1$ in Line 2. A similar approach is applied in Yuan et al. (2024) for avoid calculating the $\mathcal{L}_{\text{SFT}}$ in the future optimization steps. Then for each $\mathbf{x}_j$ in the unlabeled prompt set $\mathcal{D}_U$, $N$ response candidates $\{\mathbf{y}_i\}_{i=1}^N$ are sampled in Line 5. Then reward scores of these $N$ candidates can be calculated according to Rafailov et al. (2023) by

$$r_{ij} = \beta[\log \pi_{\boldsymbol{\theta}_t}(\mathbf{y}_{ij}|\mathbf{x}_j) - \log \pi_{\boldsymbol{\theta}_0}(\mathbf{y}_{ij}|\mathbf{x}_j)] + \beta \log Z(\mathbf{x}_j), \tag{3.9}$$

where we use the initial model parameter $\boldsymbol{\theta}_0$ as the reference policy $\pi_{\text{ref}}$. Since $\beta \geq 0$ and $\log Z(\mathbf{x}_j)$ is a constant across different response $\mathbf{y}_i$ for the same input prompt $\mathbf{x}_j$, we can drop these factors and calculate rewards in Line 6. Specially, when $t = 1$, the rank $K_{ij}$ is calculated based on the reference policy $\boldsymbol{\theta}_0$ itself. Thus we instead use the likelihood $r_{ij} = \log \pi_{\boldsymbol{\theta}_0}(\mathbf{y}_{ij}|\mathbf{x}_j)$ as the reward for this edge case. The rank for these $N$ candidates are therefore obtained in Line 7, where $J_{ij}$ means response $\mathbf{y}_{ij}$ is in the $J_{ij}$-th best in the preference list of $\mathbf{x}_j$.

**Consistency-Regularized Self-Rewarding.** As discussed in Eq. (3.8), a baseline model is required to measure the consistency. In the self-rewarding scenario, it is infeasible to add an external reward model as the baseline model. Fortunately, we can employ the model before last update $\boldsymbol{\theta}_{t-1}$ as the baseline model $\boldsymbol{\theta}'_t$ (i.e., last iteration's model) for evaluating the consistency of the model $\boldsymbol{\theta}$, thanks to chances provided by iterative training manner. Such a procedure helps mitigate the training error introduced in $t-1$-th step before obtaining $\boldsymbol{\theta}_t$. Considering a pair of tied preference pair $\mathbf{y}, \mathbf{y}'$ both performing well, as demonstrated in Figure 1. $P[\mathbf{y} \succ \mathbf{y}'|\mathbf{x}, \boldsymbol{\theta}_t]$ will be oscillating around

0.5 when $t$ grows due to the random noise. Otherwise $P[\mathbf{y} \succ \mathbf{y}'|\mathbf{x}, \boldsymbol{\theta}_t]$ might consistently converge to 0 or 1. Due to this oscillation, the consistency between $\boldsymbol{\theta}_{t-1}$ and $\boldsymbol{\theta}_t$ on this specific preference pair would be low, and the algorithm will learn less from this noninformative preference pair thus stabilize this oscillation.

Specifically, we calculate the rank of these $N$ candidates using $\boldsymbol{\theta}_{t-1}$ in Line 9 and then use the Kendall's Tau coefficient (Kendall, 1938) denoted by

$$\tau_j = \frac{2}{N(N-1)} \sum_{1 \le i < i' \le N} \left[ \mathbb{1}\left[(J_{ij} - J_{i'j})(K_{ij} - K_{i'j}) > 0\right] - \mathbb{1}\left[(J_{ij} - J_{i'j})(K_{ij} - K_{i'j}) < 0\right] \right]. \tag{3.10}$$

Kendall's Tau coefficient is a widely used coefficient (McLeod, 2005; Abdi, 2007) to measure the consistency of two ranking sequences. Basically, when two sequences perfectly aligns, $\tau_j = 1$ and when two sequence never aligns, $\tau_j = -1$. The following lemma draws the further connection between the Kendall's Tau and the regularization parameter $\lambda$ proposed in Section 3.2.

**Lemma 3.4.** Suppose the $N$ response candidate $\{\mathbf{y}_{ij}\}_i$ is i.i.d. given the prompt $\mathbf{x}_j$, then

$$\mathbb{E}[\tau_j] = 1 - 4\mathbb{E}_{\mathbf{y}, \mathbf{y}' \sim \pi_{\boldsymbol{\theta}_t}(\cdot|\mathbf{x}_j)} \mathbb{1}[\mathbf{y} \succ \mathbf{y}'|\mathbf{x}_j, \boldsymbol{\theta}_t] \mathbb{1}[\mathbf{y} \prec \mathbf{y}'|\mathbf{x}_j, \boldsymbol{\theta}_{t-1}] = 1 - 2\lambda,$$

where the expectation is taken over the randomness of sampling the $N$ candidate set.

Given Lemma 3.4, we can recover $\mathcal{C}_\lambda \approx 1 - \lambda = (1 + \tau_j)/2$ and we average all $\tau_j$ for all $\mathbf{x}_j \in \mathcal{D}_U$ in Line 11. Finally, in Line 12, we compose the preference dataset by selecting the best response $\mathbf{y}_j^+ = \mathbf{y}_{i^+j}$ and the worst response $\mathbf{y}_j^- = \mathbf{y}_{i^-j}$ which is similar with (Yuan et al., 2024).

$$\mathcal{D}_{\text{DPO}} = \{(\mathbf{x}_j, \mathbf{y}_{i^+j}, \mathbf{y}_{i^-j})|\mathbf{x}_j \in \mathcal{D}_U, i^+ = \arg\min_i J_{ij}, i^- = \arg\max_i J_{ij}\} \tag{3.11}$$

Following Theorem 3.3, we also prepare the reverse DPO dataset by switching the best response and the worst response by

$$\mathcal{D}_{\text{RDPO}} = \{(\mathbf{x}_j, \mathbf{y}_{i^-j}, \mathbf{y}_{i^+j})|\mathbf{x}_j \in \mathcal{D}_U, i^+ = \arg\min_i J_{ij}, i^- = \arg\max_i J_{ij}\} \tag{3.12}$$

and update $\boldsymbol{\theta}_{t+1}$ by minimizing the empirical loss of Eq. (3.7) in Line 14. The detailed proof of theorems and lemmas are provided in the Appendix B.

# 4 EXPERIMENT

## 4.1 EXPERIMENTAL SETUP

**Data.** In our experiments, we use Open Assistant dataset (Köpf et al., 2024) and only reserve about 3.4K human-annotated examples as the seed SFT data $\mathcal{D}_S$. To construct the unlabeled prompt dataset $\mathcal{D}_U$, we mix prompts of $\mathcal{D}_S$ with the train split of each downstream task (only reserve the prompts) including (1) ARC-Easy/Challenge (Clark et al., 2018), (2) OpenBookQA (Mihaylov et al., 2018), (3) SIQA (Sap et al., 2019), and (4) GSM8K (Cobbe et al., 2021). Finally, this process results in a total of 21K prompts in $\mathcal{D}_U$, which we distribute equally across iterative self-rewarding trainings.

**Models.** Due to limited computational resources, we mainly conduct experiments with two LLMs with about 7B parameters, including Llama-3 (Dubey et al., 2024) and Llama-2 (Touvron et al., 2023), both of which are widely used. Note that the proposed framework is designed for LLMs of any size, while validating our findings on other LLMs will be our future work.

**Baseline Methods.** We mainly compare our method with SRLM (Yuan et al., 2024) which uses the same LLM to serve as both the policy and reward model to generate preference data for iterative training. Additionally, we introduce a variant of RL with AI feedback (Guo et al., 2024a), referred to as "Oracle". In this variant, the reward model in SRLM is replaced with an external reward model to demonstrate the upper bound performance of SRLM. Specifically, we use InternLM2 (Cai et al., 2024), a specialized trained reward model, to provide the reward scores for the generated responses. We further enhance Oracle's rewarding by leveraging the labels from downstream tasks to improve the rewarding accuracy. To compare the regularization method, we introduce "SRLM + KL" variant which adds a simple regularization term $\lambda[\frac{\pi_\theta(y|x)}{\pi_{\text{ref}}(y|x)} - \frac{\pi_\theta(y'|x)}{\pi_{\text{ref}}(y'|x)}]^2$ to the original DPO

Table 1: Main results of each method on test sets of downstream tasks. The exact match accuracies are reported. The "↑" and "↓" indicate the performance improvement and degradation compared to the method's last iteration (e.g., M1 → M2 and M2 → M3), respectively. The best performance among methods using **self-rewarding** is highlighted in **bold**.

| Model | Method | Reward | Iteration | Arc-Easy | Arc-Challenge | OpenBookQA | SIQA | GSM8K | Average |
|---|---|---|---|---|---|---|---|---|---|
| Llama-3 | Initial | - | M0 | 86.29 | 80.37 | 86.00 | 68.58 | 78.01 | 79.85 |
| | SFT | - | M1 | 86.78 | 80.14 | 86.40 | 69.50 | 78.39 | 80.24 |
| | Oracle | External | M2 | 89.60↑ | 82.17↑ | 90.00↑ | 72.88↑ | 80.82↑ | 83.09↑ |
| | | | M3 | 89.31↓ | 81.31↓ | 90.20↑ | 73.75↑ | 76.04↓ | 82.12↓ |
| | SRLM | Self | M2 | 87.79↑ | 80.38↑ | 87.80↑ | 70.95↑ | 78.01↓ | 80.99↑ |
| | | | M3 | 87.17↓ | 81.23↑ | 87.30↓ | 70.37↓ | 77.48↑ | 80.71↓ |
| | SRLM + KL | | M2 | 87.92↑ | 79.78↓ | 86.60↑ | 71.49↑ | 79.38↑ | 81.03↑ |
| | | | M3 | 88.38↑ | 80.97↑ | 88.20↑ | 71.19↓ | 80.29↑ | 81.81↑ |
| | CREAM w/o RC | | M2 | 88.26↑ | 79.86↓ | 86.80↑ | 69.55↑ | 79.98↑ | 80.89↑ |
| | | | M3 | 88.09↓ | 80.55↑ | 87.20↑ | 71.39↑ | 79.23↑ | 81.29↑ |
| | CREAM | | M2 | 88.89↑ | 80.89↑ | 88.00↑ | 69.79↑ | 81.04↑ | 81.72↑ |
| | | | M3 | **89.52**↑ | **83.36**↑ | **90.20**↑ | **72.06**↑ | **81.73**↑ | **83.37**↑ |
| Llama-2 | Initial | - | M0 | 61.07 | 48.98 | 62.20 | 50.36 | 23.65 | 49.25 |
| | SFT | - | M1 | 60.44 | 48.46 | 63.20 | 50.77 | 23.88 | 49.35 |
| | Oracle | External | M2 | 70.20↑ | 55.03↑ | 75.40↑ | 63.66↑ | 30.02↑ | 58.86↑ |
| | | | M3 | 71.72↑ | 55.80↑ | 77.20↑ | 62.44↓ | 29.57↓ | 59.35↑ |
| | SRLM | Self | M2 | 58.67↓ | 46.67↓ | 59.80↑ | 49.69↓ | 25.17↑ | 48.00↓ |
| | | | M3 | 46.55↓ | 34.47↓ | 49.20↓ | 48.06↓ | 22.14↓ | 40.08↓ |
| | SRLM + KL | | M2 | 57.45↓ | 46.93↓ | 62.40↓ | 51.02↑ | 24.18↑ | 48.40↓ |
| | | | M3 | 57.20↓ | 46.42↓ | 61.60↓ | 49.08↓ | 26.08↑ | 48.08↓ |
| | CREAM w/o RC | | M2 | 58.25↓ | 45.56↓ | 61.60↓ | 51.13↑ | 24.56↑ | 48.22↓ |
| | | | M3 | 60.06↑ | **49.23**↑ | **65.40**↑ | 49.44↓ | 25.62↑ | 49.95↑ |
| | CREAM | | M2 | 58.97↓ | 47.53↓ | 62.80↓ | 50.43↓ | 24.41↑ | 48.83↓ |
| | | | M3 | **62.08**↑ | 48.81↑ | 64.60↑ | **51.22**↑ | **25.85**↑ | **50.51**↑ |

loss, where $\lambda \in [0.1, 0.2, \cdots, 1.0]$ is a hyperparameter. In order to validate the contribution of our automatically determined consistency rate via ranking correlation, we also introduce a variant "CREAM w/o RC" which replaces the ranking correlation (RC) with a manually set value as the consistency rate, where this value is searched in the range of $[0.1, 0.3, \cdots, 0.9]$.

**Implementation Details.** In our experiments, we fine-tune the initial model (M0) on the seed SFT data for 3 epochs with a learning rate of $1e-6$, resulting in model M1. Following SRLM approach, we then iteratively fine-tune the model using the preference learning objective two additional iterations, producing models M2 and M3. In the preference training of each iteration, we set $\beta = 0.1$ of DPO, and fine-tune the model for 1 epoch with a learning rate of $1e-6$. All training processes use AdamW optimizer (Loshchilov & Hutter, 2019) with a warmup ratio of $0.1$. For the response sampling stage of all SRLM methods, we use a decoding temperature of $0.8$ and generate $N = 5$ responses per prompt. For evaluating downstream tasks, we use greedy decoding to generate answers.

## 4.2 MAIN RESULTS

The main results are shown in Table 1. From these results, we have the following findings: (1) The Standard SRLM fails to achieve satisfactory performance, particularly with Llama-2 which has weaker foundation performance, which indicates its limitations for 7B-level LLMs. (2) Compared to SRLM, CREAM achieves a significant improvement, showing the advantage of introducing the proposed regularization. (3) SRLM equipped with an oracle reward model (Oracle) achieves the best performance overall. This highlights a critical challenge for all methods with unreliable rewarding. Notably, for Llama-3, CREAM even outperforms Oracle except on SIQA dataset. This superiority underlines the success of the proposed method in mitigating the rewarding bias issue. (4) The consistent improvements of CREAM across iterations show the effectiveness of the proposed regularization method in mitigating the rewarding bias issue. Results of later iterations (up to M6) in Appendix C.1 also demon-

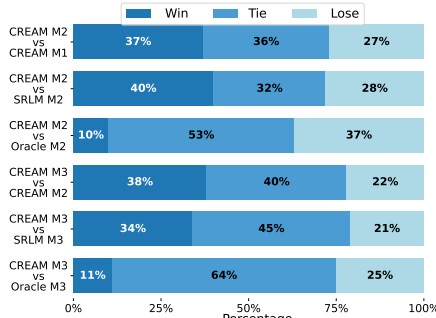

Figure 3: Arena of CREAM v.s. SRLM and Oracle, judged by GPT-4o.

strates that adding such regularization can help prevent the model from degeneration in the long term. (5) Though SRLM + KL method improves performance of SRLM via regularization, its overall performance across iterations is still inferior to CREAM, indicating CREAM performs better than directly restricting the KL divergence. (6) CREAM consistently outperforms "CREAM w/o RC" which manually set the consistency rate. And using the dynamically calculated consistency rate can eliminate the costs of hyperparameter search.

**Alignment Arena.** To more directly compare the alignment performance of CREAM, we further show the Arena win-rate of our method in Figure 3. We can find that CREAM can beat baseline methods with the same iteration, which confirms the superiority of the proposed regularized self-rewarding. Also, the win and tie rates of CREAM increase with the iteration against stronger model, i.e., Oracle, indicating the consistent performance improvements across iterations.

## 4.3 ANALYSIS

### 4.3.1 ANALYSIS OF REWARDING

**Rewarding Consistency.** We first examine the consistency of rewarding of different methods using their corresponding models from the last iteration in Table 2. Here, we use the proposed Consistency Rate $\mathcal{C}$, Kendall correlation coefficient $\tau$, Spearman correlation coefficient, and TopOrder metrics to measure the consistency, where this metric evaluates whether the final paired preference data remains the same, calculated by: $\text{TopOrder}_j = \mathbb{1}\left[\arg\min J_j = \arg\min K_j\right] \cdot \mathbb{1}\left[\arg\max J_j = \arg\max K_j\right]$, where $J_j$ and $K_j$ are the rankings provided by current model and the last iteration's model, respectively. This metric assesses whether both the least preferred and most preferred responses are consistently ranked across iterations. The results confirm that SRLMs exhibit a rewarding consistency issue, indicating that the generated preference data contains noise. In contrast, CREAM keeps the ranking consistency across iterations thanks to the regularized training.

**Prompt Rewarding v.s. DPO Rewarding.** As aforementioned, 7B level LLMs struggle with generating accurate rewards when using LLM-as-a-Judge prompting due to their limited capability. Both Figure 5 and Figure 4 clearly show that prompt rewarding is not effective for SRLM with small LLMs, as the performance starts to decrease at the first iteration (M1 → M2) when

Table 2: Ranking consistency of CREAM and SRLM across iterations using Llama-3.

| Iterations | Method | Consistency $\mathcal{C}\uparrow$ | Kendall $\tau\uparrow$ | Spearman $\uparrow$ | TopOrder $\uparrow$ |
|---|---|---|---|---|---|
| M2 vs M1 | SRLM | $0.39 \pm 0.21$ | $-0.22 \pm 0.41$ | $0.36 \pm 0.24$ | $0.03 \pm 0.18$ |
| | CREAM | $\mathbf{0.73 \pm 0.18}$ | $\mathbf{0.46 \pm 0.35}$ | $\mathbf{0.77 \pm 0.19}$ | $\mathbf{0.19 \pm 0.39}$ |
| M3 vs M2 | SRLM | $0.46 \pm 0.19$ | $-0.08 \pm 0.38$ | $0.50 \pm 0.22$ | $0.12 \pm 0.33$ |
| | CREAM | $\mathbf{0.92 \pm 0.09}$ | $\mathbf{0.84 \pm 0.19}$ | $\mathbf{0.95 \pm 0.07}$ | $\mathbf{0.59 \pm 0.49}$ |

trained on the self-rewarded preference data. In contrast, the adopted DPO rewarding method can be more suitable, since it is intrinsically aligned with the model's learning objective.

**Ranking Accuracy.** The ranking accuracy is crucial for self-rewarding methods, as it directly affects the quality of the self-generated preference data for training. Figure 4 provides an intuitive comparison of the rewarding performance. The results include the ranking accuracy on self-generated preference data and the RewardBench (Lambert et al., 2024), both of which is formulated to predict the preferred one between two responses. We use the preference data obtained by self-rewarding with ground truth ranking labels, for testing the model's in-domain ranking performance. RewardBench is used to assess models' generalization beyond the training domain. CREAM consistently achieves higher ranking accuracy than baseline methods, which promises more reliable data for training. Even though the overall ranking accuracy ($\leq 70\%$) is not satisfying, the introduced consistency regularization can help mitigate the negative impacts of potentially noisy ranking annotations.

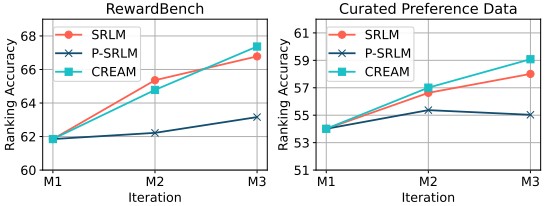

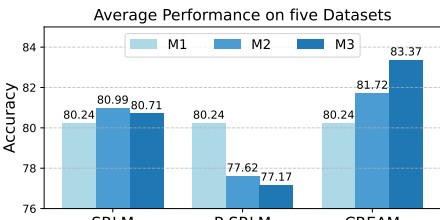

Figure 4: Pairwise ranking accuracy on RewardBench and a curated Preference Data. P-SRLM is SRLM with prompt rewarding (LLM-as-a-Judge).

Figure 5: Average performance of SRLM, P-SRLM, and CREAM. The detailed performance is shown in Table 1 and Table 13.

### 4.3.2 RELIABILITY OF SELF-CONSISTENCY

The most straightforward way to enhance the rewarding and ranking accuracy is by incorporating external reward models, such as the SRLM variant "Oracle" used in our experiments. The theoretical analysis in Eq. (3.8) suggests that we can mitigate the rewarding bias issue by calculating the ranking consistency between current model and other available baseline reward models (BRMs). However, it is not always feasible to have access to such external reward models in practice, such as

Table 3: Comparison of CREAM using oracle reward model and last iteration's model. BRM denotes the choice of baseline reward model.

| Method | BRM | Arc-E | Arc-C | OBQA | SIQA | GSM8K |
|--------|-----|-------|-------|------|------|-------|
| Llama-3 M1 | - | 86.78 | 80.14 | 86.40 | 69.50 | 78.39 |
| CREAM M2 | M0 | **88.89** | 80.89 | **88.00** | 69.79 | **81.04** |
| CREAM M2 | Oracle | 88.51 | **81.06** | 86.20 | **72.21** | 79.91 |
| Llama-2 M1 | - | 60.44 | 48.46 | 63.20 | 50.77 | 23.88 |
| CREAM M2 | M0 | 58.97 | 47.53 | 62.80 | 50.43 | **24.41** |
| CREAM M2 | Oracle | **62.42** | **48.72** | **66.00** | **51.13** | 22.52 |

the self-rewarding scenario. Thus, we instead propose to use the last iteration's model as the BRM to measure the consistency of rewarding.

**Choice of Baseline Reward Model.** To measure the impacts on CREAM of using different BRMs, we fine-tune the M1 model using CREAM with two different BRMs: the rewarding function of Oracle and the model from the last iteration (M0). As shown in Table 3, using a strong reward model as the BRM can bring better regularization effect, especially for Llama-2. However, we find that the last iteration's model also provides a reasonably reliable consistency signal for Llama-3. We attribute this to Llama-3's inherently stronger foundational alignment and better internal consistency, which allows it to effectively utilize itself without needing an external reward model.

### 4.3.3 CONSISTENCY MEASUREMENT

Besides the adopted Kendall $\tau$ coefficient, other metrics can also be used to measure the consistency between two preference ranking lists, such as Spearman coefficient (Spearman, 1904) and the aforementioned TopOrder method. We conduct a comparison experiments of using different consistency measurement methods in Table 4. We can observe that: (1) All these measurements are effective with CREAM, indicating the generalization and applica-

Table 4: Performance of CREAM using different consistency measurements with Llama-3.

| Iteration | Method | Arc-E | Arc-C | OBQA | SIQA | GSM8K |
|-----------|--------|-------|-------|------|------|-------|
| M1 | - | 86.78 | 80.14 | 86.40 | 69.50 | 78.39 |
| M2 | Spearman | 86.95 | **82.00** | 85.40 | 70.05 | 78.77 |
| | TopOrder | 87.25 | 80.12 | 86.88 | **70.83** | 79.75 |
| | **Kendall (Ours)** | **88.89** | 80.89 | **88.00** | 69.79 | **81.04** |
| M3 | Spearman | 88.76 | 81.83 | 90.00 | 70.98 | 79.15 |
| | TopOrder | 88.51 | 80.37 | 87.40 | 71.03 | 79.76 |
| | **Kendall (Ours)** | **89.52** | **83.36** | **90.20** | **72.06** | **81.73** |

bility of our regularized training approach. (2) Kendall correlation coefficient generally yields higher scores across various datasets compared to Spearman and TopOrder methods. (3) The differences in performance highlight the sensitivity of these consistency measurements. Specifically, the Spearman coefficient appears slightly less robust than Kendall's $\tau$, as analyzed in Croux & Dehon (2010). Meanwhile, TopOrder focuses only on top-1 and bottom-1 rankings, limiting its evaluation scope.

## 5 CONCLUSION

In this paper, we first formulate a generalized iterative preference fine-tuning framework for self-rewarding language models (SRLMs), which can also be applied to other iterative preference training methods. Then, we highlight the rewarding bias that arises from overconfident preference labeling, which is particularly problematic for small LLMs, such as those with 7B parameters. This rewarding bias results in the accumulation of noisy and unreliable preference data, harming the preference training and hindering alignment performance of LLMs. To address this issue, we proposed the **C**onsistency **R**egularized s**E**lf-Rewarding l**A**nguage **M**odel (CREAM), which leverages the consistency of rewarding across different iterations as a regularization signal. This approach allows the model to learn more selectively, emphasizing reliable preference data and avoiding overconfidence in preference labeling. Extensive experimental results and analysis on various natural language benchmark datasets demonstrate the effectiveness of the proposed method in mitigating the rewarding bias issue and improving the alignment performance of SRLMs. We believe that these findings can provide valuable insights for future research on self-improvement methods of LLM alignment.

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

## A    LIMITATIONS

In this section, we discuss the limitations of our work and propose potential solutions for future research. First, our method requires full fine-tuning of the model over multiple iterations, which is both time-consuming and computationally intensive. For future research, we plan to explore Parameter-Efficient Fine-Tuning (PEFT) methods to reduce training costs. Second, our approach primarily focuses on small (7B level) language models, which may limit its applicability. However, the proposed method is designed to be general and can be applied to models of any size. Third, it is worth investigating more complex consistency-regularized self-rewarding scenarios, such as how to assign different weights to various models during the self-rewarding process based on their importance, and enable these models to collaborate to improve the rewarding accuracy.

## B    PROOF OF THE THEOREMS AND LEMMAS

### B.1    PROOF OF THEOREM 3.1

*Proof of Theorem 3.1.* We denote iteration of the two-step algorithm as $t$. The algorithm starts from $(\boldsymbol{\theta}_t, z_t)$, and obtains $z_{t+1} = z_{\boldsymbol{\theta}_t}$ according to Eq. (3.3) in the preference-labeling step and then obtains $\boldsymbol{\theta}_{t+1}$ through the learning step. Since $z_{t+1} = \arg\min_z \mathcal{L}_{\text{DPO}}(\boldsymbol{\theta}_t; \mathbf{y}, \mathbf{y}', \mathbf{x}, z)$ for any $\mathbf{y}, \mathbf{y}', \mathbf{x}$ according to Eq. (3.3), we have that

$$\mathcal{L}(\boldsymbol{\theta}_t, z_{t+1}) \leq \mathcal{L}(\boldsymbol{\theta}_t, z_t). \tag{B.1}$$

And the learning step suggests that $\boldsymbol{\theta}_{t+1} = \arg\min_{\boldsymbol{\theta}} \mathcal{L}(\boldsymbol{\theta}, z_{t+1})$, yielding that

$$\mathcal{L}(\boldsymbol{\theta}_{t+1}, z_{t+1}) \leq \mathcal{L}(\boldsymbol{\theta}_t, z_{t+1}). \tag{B.2}$$

Connecting Eq. (B.1) with Eq. (B.2) yields that the loss function $\mathcal{L}(\boldsymbol{\theta}, z)$ is monotonically decreasing, i.e.

$$\cdots \leq \mathcal{L}(\boldsymbol{\theta}_{t+1}, z_{t+1}) \leq \mathcal{L}(\boldsymbol{\theta}_t, z_{t+1}) \leq \mathcal{L}(\boldsymbol{\theta}_t, z_t) \leq \cdots . \tag{B.3}$$

Since $\mathcal{L}(\boldsymbol{\theta}, z)$ is upper bounded by 0, it suggests that the sequence of $\mathcal{L}(\boldsymbol{\theta}_t, z_t)$ will converge w.r.t. the growth of $t$. □

### B.2    PROOF OF LEMMA 3.2

*Proof of Lemma 3.2.* We start by expanding $\mathbb{E}_{\mathbf{y},\mathbf{y}'\sim\pi_{\boldsymbol{\theta}_t}(\cdot|\mathbf{x})}\mathcal{L}_{\text{reg}}(\boldsymbol{\theta}; \mathbf{y}, \mathbf{y}', \mathbf{x})$ as

$$
\begin{aligned}
\mathbb{E}_{\mathbf{y},\mathbf{y}'\sim\pi_{\boldsymbol{\theta}_t}(\cdot|\mathbf{x})}\mathcal{L}_{\text{reg}}(\boldsymbol{\theta}; \mathbf{y}, \mathbf{y}', \mathbf{x}) &= \mathbb{E}_{\mathbf{y},\mathbf{y}'\sim\pi_{\boldsymbol{\theta}_t}(\cdot|\mathbf{x})}\left[\log P_{\boldsymbol{\theta}}(z=1) + \log P_{\boldsymbol{\theta}}(z=0)\right] \\
&= \mathbb{E}_{\mathbf{y},\mathbf{y}'\sim\pi_{\boldsymbol{\theta}_t}(\cdot|\mathbf{x}),\mathbf{y}\prec\mathbf{y}'}\left[\log P_{\boldsymbol{\theta}}(z=1) + \log P_{\boldsymbol{\theta}}(z=0)\right] \\
&\quad + \mathbb{E}_{\mathbf{y},\mathbf{y}'\sim\pi_{\boldsymbol{\theta}_t}(\cdot|\mathbf{x}),\mathbf{y}\succeq\mathbf{y}'}\left[\log P_{\boldsymbol{\theta}}(z=1) + \log P_{\boldsymbol{\theta}}(z=0)\right] \\
&= \mathbb{E}_{\mathbf{y},\mathbf{y}'\sim\pi_{\boldsymbol{\theta}_t}(\cdot|\mathbf{x})}P_{\boldsymbol{\theta}_t}(z=0)\left[\log P_{\boldsymbol{\theta}}(z=1) + \log P_{\boldsymbol{\theta}}(z=0)\right] \\
&\quad + \mathbb{E}_{\mathbf{y},\mathbf{y}'\sim\pi_{\boldsymbol{\theta}_t}(\cdot|\mathbf{x})}P_{\boldsymbol{\theta}_t}(z=1)\left[\log P_{\boldsymbol{\theta}}(z=1) + \log P_{\boldsymbol{\theta}}(z=0)\right],
\end{aligned}
\tag{B.4}
$$

where the second equation decompose the expectation $\mathbb{E}_{\mathbf{y},\mathbf{y}'\sim\pi_{\boldsymbol{\theta}_t}(\cdot|\mathbf{x})}$ into two expectation $\mathbb{E}_{\mathbf{y},\mathbf{y}'\sim\pi_{\boldsymbol{\theta}_t}(\cdot|\mathbf{x})} + \mathbb{E}_{\mathbf{y},\mathbf{y}'\sim\pi_{\boldsymbol{\theta}_t}(\cdot|\mathbf{x})}$, the third equation extract the event $\mathbf{y} \geq \mathbf{y}'$ as distribution $P_{\boldsymbol{\theta}_t}(z)$. Then Eq. (B.4) can be further written by

$$
\begin{aligned}
\mathbb{E}_{\mathbf{y},\mathbf{y}'\sim\pi_{\boldsymbol{\theta}_t}(\cdot|\mathbf{x})}\mathcal{L}_{\text{reg}}(\boldsymbol{\theta}; \mathbf{y}, \mathbf{y}', \mathbf{x}) &= P_{\boldsymbol{\theta}_t}(z=1)\left[\log P_{\boldsymbol{\theta}}(z=1) + \log P_{\boldsymbol{\theta}}(z=0)\right] \\
&\quad + P_{\boldsymbol{\theta}_t}(z=0)\left[\log P_{\boldsymbol{\theta}}(z=1) + \log P_{\boldsymbol{\theta}}(z=0)\right],
\end{aligned}
\tag{B.5}
$$

since both $\mathbf{y}, \mathbf{y}'$ are generated from $\pi_{\boldsymbol{\theta}_t}(\cdot|\mathbf{x})$, $P_{\boldsymbol{\theta}_t}(z=0) = P_{\boldsymbol{\theta}_t}(z=1) = 0.5$. Thus Eq. (B.5) becomes

$$\mathbb{E}_{\mathbf{y},\mathbf{y}'\sim\pi_{\boldsymbol{\theta}_t}(\cdot|\mathbf{x})}\mathcal{L}_{\text{reg}}(\boldsymbol{\theta}; \mathbf{y}, \mathbf{y}', \mathbf{x}) = 2\text{KL}(P_{\boldsymbol{\theta}}(z) \parallel P_{\boldsymbol{\theta}_t}(z)) = 2\text{KL}(P_{\boldsymbol{\theta}}(z) \parallel u(z)), \tag{B.6}$$

where $u(z)$ is the uniform binary distribution with $u(z=0) = u(z=1) = 0.5$. □

### B.3 PROOF THEOREM 3.3

*Proof of Theorem 3.3.* We start by writing down each components in $\mathcal{L}(\boldsymbol{\theta}, z)$ defined in Eq. (3.1) by

$$
\begin{aligned}
\mathcal{L}(\boldsymbol{\theta}, z) &= \mathcal{L}_{\text{SFT}}(\boldsymbol{\theta}; \mathcal{D}_{\text{S}}) + \mathbb{E}_{\mathbf{x}\sim\mathcal{D}';\mathbf{y},\mathbf{y}'\sim\pi_{\boldsymbol{\theta}_t}(\cdot|\mathbf{x})}[\mathcal{L}_{\text{DPO}}(\boldsymbol{\theta}; \mathbf{y}, \mathbf{y}', \mathbf{x}, z) + \lambda\mathcal{L}_{\text{Reg}}(\boldsymbol{\theta}; \mathbf{y}, \mathbf{y}', \mathbf{x})] \\
&= \mathcal{L}_{\text{SFT}}(\boldsymbol{\theta}; \mathcal{D}_{\text{S}}) \\
&\quad + \mathbb{E}_{\mathbf{x}\sim\mathcal{D}';\mathbf{y},\mathbf{y}'\sim\pi_{\boldsymbol{\theta}_t}(\cdot|\mathbf{x})}\left[-z(\mathbf{y}, \mathbf{y}', \mathbf{x})\log\sigma\left(\log\left(\frac{\pi_{\boldsymbol{\theta}}(\mathbf{y}|\mathbf{x})}{\pi_{\text{ref}}(\mathbf{y}|\mathbf{x})}\right) - \log\left(\frac{\pi_{\boldsymbol{\theta}}(\mathbf{y}'|\mathbf{x})}{\pi_{\text{ref}}(\mathbf{y}'|\mathbf{x})}\right)\right)\right. \\
&\quad\quad -(1 - z(\mathbf{y}, \mathbf{y}', \mathbf{x}))\log\sigma\left(\log\left(\frac{\pi_{\boldsymbol{\theta}}(\mathbf{y}'|\mathbf{x})}{\pi_{\text{ref}}(\mathbf{y}'|\mathbf{x})}\right) - \log\left(\frac{\pi_{\boldsymbol{\theta}}(\mathbf{y}|\mathbf{x})}{\pi_{\text{ref}}(\mathbf{y}|\mathbf{x})}\right)\right) \\
&\quad\quad -\lambda\log\sigma\left(\log\left(\frac{\pi_{\boldsymbol{\theta}}(\mathbf{y}|\mathbf{x})}{\pi_{\text{ref}}(\mathbf{y}|\mathbf{x})}\right) - \log\left(\frac{\pi_{\boldsymbol{\theta}}(\mathbf{y}'|\mathbf{x})}{\pi_{\text{ref}}(\mathbf{y}'|\mathbf{x})}\right)\right) \\
&\quad\quad \left.-\lambda\log\sigma\left(\log\left(\frac{\pi_{\boldsymbol{\theta}}(\mathbf{y}'|\mathbf{x})}{\pi_{\text{ref}}(\mathbf{y}'|\mathbf{x})}\right) - \log\left(\frac{\pi_{\boldsymbol{\theta}}(\mathbf{y}|\mathbf{x})}{\pi_{\text{ref}}(\mathbf{y}|\mathbf{x})}\right)\right)\right] \\
&= \mathcal{L}_{\text{SFT}}(\boldsymbol{\theta}; \mathcal{D}_{\text{S}}) \\
&\quad + \mathbb{E}_{\mathbf{x}\sim\mathcal{D}';\mathbf{y},\mathbf{y}'\sim\pi_{\boldsymbol{\theta}_t}(\cdot|\mathbf{x})}\left[-(\lambda + z(\mathbf{y}, \mathbf{y}', \mathbf{x}))\log\sigma\left(\log\left(\frac{\pi_{\boldsymbol{\theta}}(\mathbf{y}|\mathbf{x})}{\pi_{\text{ref}}(\mathbf{y}|\mathbf{x})}\right) - \log\left(\frac{\pi_{\boldsymbol{\theta}}(\mathbf{y}'|\mathbf{x})}{\pi_{\text{ref}}(\mathbf{y}'|\mathbf{x})}\right)\right)\right. \\
&\quad\quad \left.-(1 + \lambda - z(\mathbf{y}, \mathbf{y}', \mathbf{x}))\log\sigma\left(\log\left(\frac{\pi_{\boldsymbol{\theta}}(\mathbf{y}'|\mathbf{x})}{\pi_{\text{ref}}(\mathbf{y}'|\mathbf{x})}\right) - \log\left(\frac{\pi_{\boldsymbol{\theta}}(\mathbf{y}|\mathbf{x})}{\pi_{\text{ref}}(\mathbf{y}|\mathbf{x})}\right)\right)\right]
\end{aligned}
$$

where the third equation absorbs the regularization into the DPO loss. Noticing that $\lambda + z(\mathbf{y}, \mathbf{y}', \mathbf{x}) + (1 + \lambda - z(\mathbf{y}, \mathbf{y}', \mathbf{x})) = 1 + 2\lambda$, by dividing $(1 + 2\lambda)$ we have

$$
\begin{aligned}
\frac{\mathcal{L}(\boldsymbol{\theta}, z)}{1 + 2\lambda} &= \frac{\mathcal{L}_{\text{SFT}}(\boldsymbol{\theta}; \mathcal{D}_{\text{SFT}})}{1 + 2\lambda} \\
&\quad + \mathbb{E}_{\mathbf{x}\sim\mathcal{D}';\mathbf{y},\mathbf{y}'\sim\pi_{\boldsymbol{\theta}_t}(\cdot|\mathbf{x})}\left[-\frac{\lambda + z(\mathbf{y}, \mathbf{y}', \mathbf{x})}{1 + 2\lambda}\log\sigma\left(\log\left(\frac{\pi_{\boldsymbol{\theta}}(\mathbf{y}|\mathbf{x})}{\pi_{\text{ref}}(\mathbf{y}|\mathbf{x})}\right) - \log\left(\frac{\pi_{\boldsymbol{\theta}}(\mathbf{y}'|\mathbf{x})}{\pi_{\text{ref}}(\mathbf{y}'|\mathbf{x})}\right)\right)\right. \\
&\quad\quad \left.-\frac{1 + \lambda - z(\mathbf{y}, \mathbf{y}', \mathbf{x})}{1 + 2\lambda}\log\sigma\left(\log\left(\frac{\pi_{\boldsymbol{\theta}}(\mathbf{y}'|\mathbf{x})}{\pi_{\text{ref}}(\mathbf{y}'|\mathbf{x})}\right) - \log\left(\frac{\pi_{\boldsymbol{\theta}}(\mathbf{y}|\mathbf{x})}{\pi_{\text{ref}}(\mathbf{y}|\mathbf{x})}\right)\right)\right].
\end{aligned}
$$

When $z(\mathbf{y}, \mathbf{y}'\mathbf{x}) = 1$, $(\lambda + z(\mathbf{y}, \mathbf{y}', \mathbf{x}))/(1 + 2\lambda) = 1 - \lambda/(1 + 2\lambda)$ and $(1 + \lambda - z(\mathbf{y}, \mathbf{y}', \mathbf{x}))(1 + 2\lambda) = \lambda/(1 + 2\lambda)$. Therefore, letting $\mathcal{C}_\lambda = \lambda/(1 + 2\lambda)$ yields that

$$
\begin{aligned}
\frac{\mathcal{L}(\boldsymbol{\theta}, z)}{1 + 2\lambda} &= \frac{\mathcal{L}_{\text{SFT}}(\boldsymbol{\theta}; \mathcal{D}_{\text{SFT}})}{1 + 2\lambda} \\
&\quad + \mathbb{E}_{\mathbf{x}\sim\mathcal{D}';\mathbf{y},\mathbf{y}'\sim\pi_{\boldsymbol{\theta}_t}(\cdot|\mathbf{x})}[(1 - \mathcal{C}_\lambda)\mathcal{L}_{\text{DPO}}(\boldsymbol{\theta}; \mathbf{y}, \mathbf{y}', \mathbf{x}, z) + \mathcal{C}_\lambda\mathcal{L}_{\text{DPO}}(\boldsymbol{\theta}; \mathbf{y}, \mathbf{y}', \mathbf{x}, 1 - z)],
\end{aligned}
$$

which completes the proof since minimizing $\mathcal{L}(\boldsymbol{\theta}, z)/(1 + \lambda)$ is equivalent with minimizing $\mathcal{L}(\boldsymbol{\theta}, z)$ itself. $\qquad\square$

### B.4 PROOF OF LEMMA 3.4

*Proof of Lemma 3.4.* To begin with, according to the ranking of $J_{ij}$, the sufficient and necessary condition for $J_{ij} - J_{i'j} > 0$ is that $r_{ij} < r_{i'j}$. Similarly, the sufficient and necessary condition for $K_{ij} > K_{i'j}$ is that $r'_{ij} < r'_{i'j}$. As a result, the indicator becomes

$$
\mathbb{1}[(J_{ij} - J_{i'j})(K_{ij} - K_{i'j}) > 0] = \mathbb{1}[(r_{ij} - r_{i'j})(r'_{ij} - r'_{i'j}) > 0] \tag{B.7}
$$

$$
\mathbb{1}[(J_{ij} - J_{i'j})(K_{ij} - K_{i'j}) < 0] = \mathbb{1}[(r_{ij} - r_{ij'})(r'_{ij} - r'_{i'j}) < 0]. \tag{B.8}
$$

Since $r_{ij} > r_{i'j}$ yields $\mathbf{y}_{ij} \succ \mathbf{y}_{i'j}$ under the input prompt $xb_j$ and language model $\boldsymbol{\theta}_t$, Eq. (B.7) becomes

$$
\begin{aligned}
\mathbb{1}[(J_{ij} - J_{i'j})(K_{ij} - K_{i'j}) > 0] &= \mathbb{1}[r_{ij} > r_{i'j}]\,\mathbb{1}[r'_{ij} > r'_{i'j}] + \mathbb{1}[r_{ij} < r_{i'j}]\,\mathbb{1}[r'_{ij} < r'_{i'j}] \\
&= \mathbb{1}[\mathbf{y}_{ij} \succ \mathbf{y}_{i'j}|\mathbf{x}_j, \boldsymbol{\theta}_t]\,\mathbb{1}[\mathbf{y}_{ij} \succ \mathbf{y}_{i'j}|\mathbf{x}_j, \boldsymbol{\theta}_{t-1}] \\
&\quad + \mathbb{1}[\mathbf{y}_{ij} \prec \mathbf{y}_{i'j}|\mathbf{x}_j, \boldsymbol{\theta}_t]\,\mathbb{1}[\mathbf{y}_{ij} \prec \mathbf{y}_{i'j}|\mathbf{x}_j, \boldsymbol{\theta}_{t-1}]. \tag{B.9}
\end{aligned}
$$

As a result, since $\mathbf{y}_{ij}$ are i.i.d. given $\mathbf{x}_j$, the expectation of first part of the Kendall's Tau coefficient is

$$
\begin{aligned}
&\mathbb{E}[\mathbb{1}[(J_{ij} - J_{i'j})(K_{ij} - K_{i'j}) > 0]] - \mathbb{E}[\mathbb{1}[(J_{ij} - J_{i'j})(K_{ij} - K_{i'j}) < 0]] \\
&= \mathbb{E}_{\mathbf{y}_{ij},\mathbf{y}_{i'j} \sim \pi_{\boldsymbol{\theta}_t}(\cdot|\mathbf{x}_j)}[\mathbb{1}[\mathbf{y}_{ij} \succ \mathbf{y}_{i'j}|\mathbf{x}_j, \boldsymbol{\theta}_t]\, \mathbb{1}[\mathbf{y}_{ij} \succ \mathbf{y}_{i'j}|\mathbf{x}_j, \boldsymbol{\theta}_{t-1}]] \\
&\quad + \mathbb{E}_{\mathbf{y}_{ij},\mathbf{y}_{i'j} \sim \pi_{\boldsymbol{\theta}_t}(\cdot|\mathbf{x}_j)}[\mathbb{1}[\mathbf{y}_{ij} \prec \mathbf{y}_{i'j}|\mathbf{x}_j, \boldsymbol{\theta}_t]\, \mathbb{1}[\mathbf{y}_{ij} \prec \mathbf{y}_{i'j}|\mathbf{x}_j, \boldsymbol{\theta}_{t-1}]] \\
&\quad - \mathbb{E}_{\mathbf{y}_{ij},\mathbf{y}_{i'j} \sim \pi_{\boldsymbol{\theta}_t}(\cdot|\mathbf{x}_j)}[\mathbb{1}[\mathbf{y}_{ij} \prec \mathbf{y}_{i'j}|\mathbf{x}_j, \boldsymbol{\theta}_t]\, \mathbb{1}[\mathbf{y}_{ij} \succ \mathbf{y}_{i'j}|\mathbf{x}_j, \boldsymbol{\theta}_{t-1}]] \\
&\quad - \mathbb{E}_{\mathbf{y}_{ij},\mathbf{y}_{i'j} \sim \pi_{\boldsymbol{\theta}_t}(\cdot|\mathbf{x}_j)}[\mathbb{1}[\mathbf{y}_{ij} \succ \mathbf{y}_{i'j}|\mathbf{x}_j, \boldsymbol{\theta}_t]\, \mathbb{1}[\mathbf{y}_{ij} \prec \mathbf{y}_{i'j}|\mathbf{x}_j, \boldsymbol{\theta}_{t-1}]] \\
&= \mathbb{E}_{\mathbf{y}_{ij},\mathbf{y}_{i'j} \sim \pi_{\boldsymbol{\theta}_t}(\cdot|\mathbf{x}_j)}[\mathbb{1}[\mathbf{y}_{ij} \succ \mathbf{y}_{i'j}|\mathbf{x}_j, \boldsymbol{\theta}_t](\mathbb{1}[\mathbf{y}_{ij} \succ \mathbf{y}_{i'j}|\mathbf{x}_j, \boldsymbol{\theta}_{t-1}] - \mathbb{1}[\mathbf{y}_{ij} \prec \mathbf{y}_{i'j}|\mathbf{x}_j, \boldsymbol{\theta}_{t-1}])] \\
&\quad + \mathbb{E}_{\mathbf{y}_{ij},\mathbf{y}_{i'j} \sim \pi_{\boldsymbol{\theta}_t}(\cdot|\mathbf{x}_j)}[\mathbb{1}[\mathbf{y}_{ij} \prec \mathbf{y}_{i'j}|\mathbf{x}_j, \boldsymbol{\theta}_t](\mathbb{1}[\mathbf{y}_{ij} \prec \mathbf{y}_{i'j}|\mathbf{x}_j, \boldsymbol{\theta}_{t-1}] - \mathbb{1}[\mathbf{y}_{ij} \succ \mathbf{y}_{i'j}|\mathbf{x}_j, \boldsymbol{\theta}_{t-1}])] \\
&= \mathbb{E}_{\mathbf{y}_{ij},\mathbf{y}_{i'j} \sim \pi_{\boldsymbol{\theta}_t}(\cdot|\mathbf{x}_j)}[\mathbb{1}[\mathbf{y}_{ij} \succ \mathbf{y}_{i'j}|\mathbf{x}_j, \boldsymbol{\theta}_t](1 - 2\,\mathbb{1}[\mathbf{y}_{ij} \prec \mathbf{y}_{i'j}|\mathbf{x}_j, \boldsymbol{\theta}_{t-1}])] \\
&\quad - \mathbb{E}_{\mathbf{y}_{ij},\mathbf{y}_{i'j} \sim \pi_{\boldsymbol{\theta}_t}(\cdot|\mathbf{x}_j)}[\mathbb{1}[\mathbf{y}_{ij} \prec \mathbf{y}_{i'j}|\mathbf{x}_j, \boldsymbol{\theta}_t](1 - 2\,\mathbb{1}[\mathbf{y}_{ij} \prec \mathbf{y}_{i'j}|\mathbf{x}_j, \boldsymbol{\theta}_{t-1}])] \\
&= \mathbb{E}_{\mathbf{y}_{ij},\mathbf{y}_{i'j} \sim \pi_{\boldsymbol{\theta}_t}(\cdot|\mathbf{x}_j)}[\mathbb{1}[\mathbf{y}_{ij} \succ \mathbf{y}_{i'j}|\mathbf{x}_j, \boldsymbol{\theta}_t] - \mathbb{1}[\mathbf{y}_{ij} \prec \mathbf{y}_{i'j}|\mathbf{x}_j, \boldsymbol{\theta}_t]] \\
&\quad + 2\mathbb{E}_{\mathbf{y}_{ij},\mathbf{y}_{i'j} \sim \pi_{\boldsymbol{\theta}_t}(\cdot|\mathbf{x}_j)}[\mathbb{1}[\mathbf{y}_{ij} \prec \mathbf{y}_{i'j}|\mathbf{x}_j, \boldsymbol{\theta}_{t-1}](\mathbb{1}[\mathbf{y}_{ij} \prec \mathbf{y}_{i'j}|\mathbf{x}_j, \boldsymbol{\theta}_t] - \mathbb{1}[\mathbf{y}_{ij} \succ \mathbf{y}_{i'j}|\mathbf{x}_j, \boldsymbol{\theta}_t])] \\
&= 0 + 2\mathbb{E}_{\mathbf{y}_{ij},\mathbf{y}_{i'j} \sim \pi_{\boldsymbol{\theta}_t}(\cdot|\mathbf{x}_j)}[\mathbb{1}[\mathbf{y}_{ij} \prec \mathbf{y}_{i'j}|\mathbf{x}_j, \boldsymbol{\theta}_{t-1}](1 - 2\,\mathbb{1}[\mathbf{y}_{ij} \succ \mathbf{y}_{i'j}|\mathbf{x}_j, \boldsymbol{\theta}_t])] \quad\quad (B.10)
\end{aligned}
$$

where the second equation merge the terms together, and the third equation is due to the fact $\mathbb{1}[\mathbf{y}_{ij} \prec \mathbf{y}_{i'j}] + \mathbb{1}[\mathbf{y}_{ij} \succ \mathbf{y}_{i'j}] = 1$, the forth equation reorganize the term and the fifth equation is due to the fact that $\mathbb{E}_{\mathbf{y}_{ij},\mathbf{y}_{i'j} \sim \pi_{\boldsymbol{\theta}_t}(\cdot|\mathbf{x}_j)}[\mathbb{1}[\mathbf{y}_{ij} \succ \mathbf{y}_{i'j}|\mathbf{x}_j, \boldsymbol{\theta}_t] - \mathbb{1}[\mathbf{y}_{ij} \prec \mathbf{y}_{i'j}|\mathbf{x}_j, \boldsymbol{\theta}_t]] = 0$ due to symmetry. Similarly by reversing the $\prec$ and $\succ$, we can write Eq. (B.10) by

$$
\begin{aligned}
&\mathbb{E}[\mathbb{1}[(J_{ij} - J_{i'j})(K_{ij} - K_{i'j}) > 0]] - \mathbb{E}[\mathbb{1}[(J_{ij} - J_{i'j})(K_{ij} - K_{i'j}) < 0]] \\
&= 0 + 2\mathbb{E}_{\mathbf{y}_{ij},\mathbf{y}_{i'j} \sim \pi_{\boldsymbol{\theta}_t}(\cdot|\mathbf{x}_j)}[\mathbb{1}[\mathbf{y}_{ij} \succ \mathbf{y}_{i'j}|\mathbf{x}_j, \boldsymbol{\theta}_{t-1}](1 - 2\,\mathbb{1}[\mathbf{y}_{ij} \prec \mathbf{y}_{i'j}|\mathbf{x}_j, \boldsymbol{\theta}_t])]. \quad\quad (B.11)
\end{aligned}
$$

Adding Eq. (B.10) and Eq. (B.11) together yields

$$
\begin{aligned}
&2\mathbb{E}[\mathbb{1}[(J_{ij} - J_{i'j})(K_{ij} - K_{i'j}) > 0]] - \mathbb{E}[\mathbb{1}[(J_{ij} - J_{i'j})(K_{ij} - K_{i'j}) < 0]] \\
&= 2\mathbb{E}_{\mathbf{y}_{ij},\mathbf{y}_{i'j} \sim \pi_{\boldsymbol{\theta}_t}(\cdot|\mathbf{x}_j)}[\mathbb{1}[\mathbf{y}_{ij} \succ \mathbf{y}_{i'j}|\mathbf{x}_j, \boldsymbol{\theta}_{t-1}] + \mathbb{1}[\mathbf{y}_{ij} \prec \mathbf{y}_{i'j}|\mathbf{x}_j, \boldsymbol{\theta}_{t-1}]] \\
&\quad - 4\mathbb{E}_{\mathbf{y}_{ij},\mathbf{y}_{i'j} \sim \pi_{\boldsymbol{\theta}_t}(\cdot|\mathbf{x}_j)}[\mathbb{1}[\mathbf{y}_{ij} \succ \mathbf{y}_{i'j}|\mathbf{x}_j, \boldsymbol{\theta}_t]\, \mathbb{1}[\mathbf{y}_{ij} \prec \mathbf{y}_{i'j}|\mathbf{x}_j, \boldsymbol{\theta}_{t-1}]] \\
&\quad - 4\mathbb{E}_{\mathbf{y}_{ij},\mathbf{y}_{i'j} \sim \pi_{\boldsymbol{\theta}_t}(\cdot|\mathbf{x}_j)}[\mathbb{1}[\mathbf{y}_{ij} \prec \mathbf{y}_{i'j}|\mathbf{x}_j, \boldsymbol{\theta}_t]\, \mathbb{1}[\mathbf{y}_{ij} \succ \mathbf{y}_{i'j}|\mathbf{x}_j, \boldsymbol{\theta}_{t-1}]] \\
&= 2 - 8\mathbb{E}_{\mathbf{y}_{ij},\mathbf{y}_{i'j} \sim \pi_{\boldsymbol{\theta}_t}(\cdot|\mathbf{x}_j)}[\mathbb{1}[\mathbf{y}_{ij} \succ \mathbf{y}_{i'j}|\mathbf{x}_j, \boldsymbol{\theta}_t]\, \mathbb{1}[\mathbf{y}_{ij} \prec \mathbf{y}_{i'j}|\mathbf{x}_j, \boldsymbol{\theta}_{t-1}]], \quad\quad (B.12)
\end{aligned}
$$

where the final equation is because $\mathbb{E}[\mathbb{1}[\mathbf{y}_{ij} \prec \mathbf{y}_{i'j}|\mathbf{x}_j, \boldsymbol{\theta}_t]\, \mathbb{1}[\mathbf{y}_{ij} \succ \mathbf{y}_{i'j}|\mathbf{x}_j, \boldsymbol{\theta}_{t-1}]] = \mathbb{E}[\mathbb{1}[\mathbf{y}_{ij} \succ \mathbf{y}_{i'j}|\mathbf{x}_j, \boldsymbol{\theta}_t]\, \mathbb{1}[\mathbf{y}_{ij} \prec \mathbf{y}_{i'j}|\mathbf{x}_j, \boldsymbol{\theta}_{t-1}]]$ due to symmetry. Divide Eq. (B.12) by 2 yields the claimed result. $\qquad\square$

## C  ADDITIONAL RESULTS

### C.1  SUSTAINABILITY OF SELF-REWARDING

To explore continuous improvements and further validate the effectiveness of the introduced regularization, we conduct additional experiments with SRLM (baseline method) and CREAM (ours) using Llama-3 for the 4th, 5th, and 6th iterations. From the results in Table 5, we have the following findings: (1) CREAM converges at the 4th iteration (M4) while the baseline method SRLM started performance degradation much earlier at the 2nd iteration (M2). This shows that CREAM helps stabilize the self-improvement training by mitigating the rewarding bias issue. (2) CREAM does not seriously harm the performance after convergence (i.e., during M5 and M6), while SRLM drastically drops the performance. This suggests that adding this consistency-based regularization is beneficial to preventing model from degeneration in the long term.

### C.2  DIFFERENCE BETWEEN CREAM LOSS AND WEIGHTED DPO LOSS

We would like to emphasize that the final form of CREAM, which combines a normal DPO and a reversed DPO, cannot simply be replaced by reducing the weight in a normal DPO (weight * DPO

Table 5: Results for SRLM and CREAM using Llama-3 for six iterations. ↑ and ↓ indicate the performance improvement and degradation compared to the method's last iteration.

| Method | Iteration | Arc-Easy | Arc-Challenge | OpenBookQA | SIQA | GSM8K | Average |
|---|---|---|---|---|---|---|---|
| Initial | M0 | 86.29 | 80.37 | 86.00 | 68.6 | 78.01 | 79.85 |
| SFT | M1 | 86.78 | 80.14 | 86.40 | 69.50 | 78.39 | 80.24 |
| SRLM | M2 | 87.79 ↑ | 80.38 ↑ | 87.80 ↑ | 70.95 ↑ | 78.01 ↓ | 80.99 ↑ |
| | M3 | 87.17 ↓ | 81.23 ↑ | 87.30 ↓ | 70.37 ↓ | 77.48 ↓ | 80.71 ↓ |
| | M4 | 86.07 ↓ | 78.33 ↓ | 87.80 ↑ | 68.58 ↓ | 75.83 ↓ | 79.32 ↓ |
| | M5 | 84.34 ↓ | 76.53 ↓ | 85.80 ↓ | 66.84 ↓ | 64.22 ↓ | 75.55 ↓ |
| | M6 | 76.22 ↓ | 72.36 ↓ | 76.00 ↓ | 59.06 ↓ | 59.29 ↓ | 68.59 ↓ |
| CREAM | M2 | 88.89 ↑ | 80.89 ↑ | 88.00 ↑ | 69.79 ↑ | 81.04 ↑ | 81.72 ↑ |
| | M3 | 89.52 ↑ | 83.36 ↑ | 90.20 ↑ | 72.06 ↑ | 81.73 ↑ | 83.37 ↑ |
| | M4 | 89.56 ↑ | 82.68 ↓ | 90.80 ↑ | 72.93 ↑ | 82.26 ↑ | 83.65 ↑ |
| | M5 | 89.35 ↓ | 82.08 ↓ | 90.20 ↓ | 72.06 ↓ | 81.73 ↓ | 83.08 ↓ |
| | M6 | 88.85 ↓ | 81.57 ↓ | 89.60 ↓ | 71.14 ↓ | 82.49 ↑ | 82.73 ↓ |

Loss). Specifically, the loss adopted by CREAM is $C \log \sigma(r(y_+) - r(y_-)) + (1 - C) \log \sigma(r(y_-) - r(y_+))$ differs fundamentally from a weighted DPO loss $C \log \sigma(r(y_+) - r(y_-))$. From the optimization perceptive, the latter weighted DPO loss behaves similar with a regular DPO loss with a smaller learning rate. In addition, since the preference probability $P(y_+ \prec y_-) = \sigma(r(y_+) - r(y_-))$ and $P(y_+ \succ y_-) = \sigma(r(y_-) - r(y_+))$, we would highlight that the CREAM loss can be kind of viewed as a cross-entropy loss $C \log P(y_+ \succ y_-) + (1 - C) \log P(y_- \succ y_+)$ with label-smoothing factor $C$. Thus, The weighted DPO loss cannot deliver such a observation.

**Detailed Derivation.** The loss used by our CREAM can be expressed as:

$$C \cdot D + (1 - C) \cdot RD,$$

where $C$ is the consistency rate, and $D$ and $RD$ are the DPO loss and ReverseDPO loss, respectively.

For the DPO loss, $D = \log \sigma(a - b)$, where the sigmoid function $\sigma$ converts the reward gap $(a - b)$ into a probability $P$. For $RD$, the reward gap is $(b - a)$. Recall that the sigmoid function has a property that $\sigma(x) = 1 - \sigma(-x)$. Using this property, we can convert the sigmoid (gap part) of $RD$ to $(1 - P)$. Thus, the $RD$ loss is $\log(1 - P)$, and the $D$ loss is $\log P$.

Combining these, we can derive

$$C \cdot D + (1 - C) \cdot RD = C \cdot \log P + (1 - C) \cdot \log(1 - P),$$

which corresponds to a cross-entropy loss for a binary classification task (binary preference judgment). The proposed loss acts as a label smoothing to regularize the training. This regularization is expected to enhance the model's generalization ability and performance during training.

**Gradient Analysis.** To further understand the impact of the sum of normal DPO loss and reversed DPO loss, we analyze the gradient of the adopted loss function. Recall that CREAM uses the

$$L = C \cdot D + (1 - C) \cdot RD$$

for training, where $C$ is the consistency rate, and $D$ and $RD$ are the DPO loss and ReverseDPO loss, respectively.

We can simplify the DPO loss by substituting the reward gap

$$\left[ \log \frac{\pi_\theta(y^+)}{\pi_{\text{ref}}(y^+)} - \log \frac{\pi_\theta(y^-)}{\pi_{\text{ref}}(y^-)} \right]$$

as $x_\theta$. Then, the $D$ loss is written as $\log \sigma(x_\theta)$, and the $RD$ loss is written as $\log \sigma(-x_\theta)$.

Recall that the sigmoid function $\sigma$ has the property that $\sigma(x) = 1 - \sigma(-x)$. Then, $RD$ can be converted to $\log(1 - \sigma(x_\theta))$. Our regularized loss $L$ is

$$C \log \sigma(x_\theta) + (1 - C) \log(1 - \sigma(x_\theta)).$$

Table 6: Comparison of CREAM and SRLM using weighted DPO loss on Llama-3. "NA" indicates that the method cannot generate fluent sentences.

| Method | Iteration | Arc-Easy | Arc-Challenge | OpenBookQA | SIQA | GSM8K | Average |
|---|---|---|---|---|---|---|---|
| SRLM + Weight = -1.0 | M2 | NA | NA | NA | NA | NA | NA |
| SRLM + Weight = -0.5 | M2 | NA | NA | NA | NA | NA | NA |
| SRLM + Weight = -0.25 | M2 | NA | NA | NA | NA | NA | NA |
| SRLM + Weight = 0.25 | M2 | **88.97** | 80.38 | 87.00 | **71.39** | 78.7 | 81.29 |
| SRLM + Weight = 0.50 | M2 | 86.49 | 79.61 | 87.60 | 70.42 | 79.00 | 80.62 |
| SRLM + Weight = 1.00 | M2 | 87.79 | 80.38 | 87.80 | 70.95 | 78.01 | 80.99 |
| CREAM | M2 | 88.89 | **80.89** | **88.00** | 69.79 | **81.04** | **81.72** |
| SRLM + Weight = -1.0 | M3 | 24.16 | 22.53 | 27.6 | 30.91 | 1.74 | 21.39 |
| SRLM + Weight = -0.5 | M3 | NA | NA | NA | NA | NA | NA |
| SRLM + Weight = -0.25 | M3 | NA | NA | NA | NA | NA | NA |
| SRLM + Weight = 0.25 | M3 | 88.93 | 81.40 | 89.20 | 71.34 | 75.74 | 81.32 |
| SRLM + Weight = 0.50 | M3 | 88.13 | 81.74 | 89.00 | 70.73 | 75.89 | 81.10 |
| SRLM + Weight = 1.00 | M3 | 87.17 | 81.23 | 87.30 | 70.37 | 77.48 | 80.71 |
| CREAM | M3 | **89.52** | **83.36** | **90.20** | **72.06** | **81.73** | **83.37** |

The gradient of $L$ with respect to model parameters $\theta$ is

$$\frac{\partial L}{\partial \theta} = (C - \sigma(x_\theta))\frac{\partial x_\theta}{\partial \theta}.$$

In particular, when $C = \frac{1}{2}$, it pushes the model to learn $\sigma(x_\theta) = \frac{1}{2}$, i.e., encouraging the model to have

$$\frac{\pi_\theta(y^+)}{\pi_{\text{ref}}(y^+)} = \frac{\pi_\theta(y^-)}{\pi_{\text{ref}}(y^-)}.$$

So far, there is no theoretical evidence that the likelihoods of both $y^+$ and $y^-$ will decrease. However, based on empirical results from existing works (Rafailov et al., 2023; Azar et al., 2023; Hong et al., 2024; Ethayarajh et al., 2024), the $\frac{\partial x_\theta}{\partial \theta}$ term will decrease the likelihoods of both $y^+$ and $y^-$ while increasing their gap. Thus, the combined optimization will potentially lead to: (1) a decrease in the likelihoods of both $y^+$ and $y^-$, (2) an increase in the gap between $y^+$ and $y^-$, while maintaining

$$\frac{\pi_\theta(y^+)}{\pi_{\text{ref}}(y^+)} \approx \frac{\pi_\theta(y^-)}{\pi_{\text{ref}}(y^-)}.$$

**Empirical Results.** We also take additional experiments to validate the SRLM using weight $*$ DPO Loss, where weight $\in [-1.0, 1.0]$. For SRLM + Weight method, M3 is trained based on the best checkpoint of M2 across different weights. The results are shown in Table 6. Note that "NA" means the negative of the DPO loss leads to catastrophic forgetting, where the LLM fails to generate fluent sentences. According to the results, we observe that CREAM outperforms the SRLM+weight method, indicating its effectiveness and irreplaceability.

## C.3 CONSISTENCY TREND

In CREAM, if the consistency rate $C$ reaches to 0, the preference order of the training data will be totally reversed, which may lead to overconfidence in the reversed order. However, We would like to highlight that the consistency between $\theta_{t-1}$ and $\theta_t$ can rarely be as low as $C \approx 0$. This is because $C = 0$ means for any preference pair $y, y'$, two consecutive model $\theta$ and $\theta_{t-1}$ give totally reversed preference. For the scaled Kendal Tau, that means if the ranking for 5 generated responses is $A > B > C > D > E$, to reach $C = 0$, we need to change the ranking to $E > D > C > B > A$, which is very rare in SRLM. Since $\theta_t$ is trained using the prefernece data given by $\theta_{t-1}$, they tend to have similar behavior and thus such a circumstance making $C = 0$ can hardly happen in practice.

Further, we analyze the distribution of the consistency and supply the results for 4th, 5th and 6th iterations as follows. The results in Table 7 show the consistency trend for CREAM. We can find that the initial consistency rate 0.39 is acceptable, which will not result in heavily relying on reverse preferences. Besides, the regularization of CREAM is shown to encourage the consistency of rankings and maintain stability in the later rounds of training.

Table 7: The distribution of the training samples' consistency rate $C$ in the six iterations of CREAM.

| Consistency Rate
Iteration | 0-0.2 | 0.2-0.4 | 0.4-0.6 | 0.6-0.8 | 0.8-1.0 | Total | Avg. $C$ |
|---|---|---|---|---|---|---|---|
| M1 - M2 | 1329 (13.86%) | 3129 (32.62%) | 2977 (31.04%) | 1647 (17.17%) | 510 (5.32%) | 9592 (100%) | 0.39 |
| M2 - M3 | 1 (0.01%) | 10 (0.10%) | 48 (0.50%) | 448 (4.67%) | 9085 (94.71%) | 9592 (100%) | 0.92 |
| M3 - M4 | 3 (0.03%) | 24 (0.25%) | 195 (2.03%) | 1249 (13.03%) | 8118 (84.66%) | 9589 (100%) | 0.87 |
| M4 - M5 | 2 (0.02%) | 10 (0.10%) | 121 (1.26%) | 888 (9.26%) | 8571 (89.36%) | 9592 (100%) | 0.89 |
| M5 - M6 | 6 (0.06%) | 109 (1.14%) | 573 (5.97%) | 2129 (22.20%) | 6775 (70.63%) | 9592 (100%) | 0.81 |

## C.4 COMPARISON OF ENSEMBLE METHODS

we also include an ablation study on using conservative value estimation methods (Coste et al., 2023) (**Ensemble-Worst**) and **Ensemble-Mean** in SRLMs. Specifically, ensemble method trains 3 different $\pi_\theta$ models with 3 different learning rates [7e-7, 1e-6, 3e-6] to serve as the ensemble reward models in each iteration, since no external models can be involved in the self-rewarding scenario. Then, these three models would independently reward and rank the responses:

- Ensemble-Worst selects the minimum reward for ranking the response.
- Ensemble-Mean selects the average reward for ranking the response.

Table 8 suggests that CREAM has advantage against Ensemble methods in the alignment performance a cross iterations. Additionally, compared with the ensemble model which requires a batch of reward models, CREAM only requires the model from the last one iteration and thus would be more efficient than the ensemble methods.

**Further Discussion on Robust Preference Optimization** In this paragraph, we would like to take in-depth discussion on the related work robust preference optimization (Fisch et al., 2024), in order to better understand the contribution of CREAM in the self-rewarding scenario. In details, Fisch et al. (2024) suggested injecting pessimism by ensemble or introducing the KL divergence $\text{KL}(\pi_{\theta_t} \parallel \pi_{\text{ref}})$. However, we would like to highlight that this method cannot be directly applied in our task scenario, due to the nature of the self-rewarding setting.

First, regarding the reward distillation, Fisch et al. (2024) suggested distilling the (ensembled) reward information into the language models. However, knowledge distillation usually assumes distilling information from a large model into a small model (Hinton, 2015; Buciluă et al., 2006; Mirzadeh et al., 2020). This makes perfect sense when the reward is given (e.g., from human preference, an oracle reward model, or a more advanced model). However, in SRLMs, the reward preference is given by the language model in the last iteration. The model capacity does not change over time, so the distillation may not help improve the model. For instance, if we extract $r^*$ from language model $\pi_{\theta_{t-1}}$, the reward distillation (Eq. 7 in Fisch et al. 2024) becomes

$$\theta_t = \arg\min_\theta \mathbb{E}\left(\log\frac{\pi_{\theta_{t-1}}(y_+ \mid x)}{\pi_{\theta_{t-1}}(y_- \mid x)} - \log\frac{\pi_\theta(y_+ \mid x)}{\pi_\theta(y_- \mid x)}\right)^2,$$

and a trivial minimizer is $\theta_t = \theta_{t-1}$ with loss equal to zero. This indicates that if we use the same, single model for reward distillation, the model can hardly be improved.

Second, we would like to acknowledge that the ensembled model used in Fisch et al. (2024) and other works Coste et al. (2023); Eisenstein et al. (2023); Ramé et al. (2024) can provide pessimism, especially when we have a batch of reward models. For example, Fisch et al. (2024) used 5 reward models in their Pessimistic Ensemble DPO (e-DPO) variant, which has the best performance. However, in the **self-rewarding** setting, each reward model is trained by ourselves, so obtaining a reward model would be costly.

For example, let's assume $\pi_{\text{ref}}$ is the reference model, $\pi_1$ is the model trained by SFT, $\pi_2$ is the first round of self-rewarding training, and $\pi_3$ is the second round, etc. we would see such ensemble will not work for training $\pi_2$, and for $\pi_3$, will ensemble 2 models; $\pi_4$ will ensemble 3 models. Note that though we can use different data subsets or hyperparameters to obtain multiple reward models in the earlier iterations, it can bring additional training costs. It is obvious that this algorithm needs to store all models $\pi_1, \pi_2, \ldots, \pi_M$ as the iteration progresses. In the case where more models are required to improve the ensemble performance (e.g., we need to ensemble 5 separate $\pi_1$ for

Table 8: Comparison of SRLM, Ensemble-Worst, Ensemble-Mean and CREAM on Llama-3. The best performance among methods in each iteration is highlighted in **bold**.

| Method | Iteration | Arc-Easy | Arc-Challenge | OpenBookQA | SIQA | GSM8K | Average |
|---|---|---|---|---|---|---|---|
| SRLM | M2 | 87.79 | 80.38 | 87.80 | **70.95** | 78.01 | 80.99 |
| Ensemble-Worst | M2 | 88.00 | 79.78 | 87.00 | 69.96 | 78.54 | 80.66 |
| Ensemble-Mean | M2 | 88.47 | 80.80 | **88.00** | 69.50 | 80.67 | 81.49 |
| CREAM | M2 | **88.89** | **80.89** | **88.00** | 69.79 | **81.04** | **81.72** |
| SRLM | M3 | 87.17 | 81.23 | 87.30 | 70.37 | 77.48 | 80.71 |
| Ensemble-Worst | M3 | 87.75 | 80.89 | 86.80 | 70.78 | 78.70 | 80.98 |
| Ensemble-Mean | M3 | 88.85 | 80.89 | 87.00 | 69.96 | 79.98 | 81.34 |
| CREAM | M3 | **89.52** | **83.36** | **90.20** | **72.06** | **81.73** | **83.37** |

Pessimistic Ensemble DPO), this storage and computation consumption increase significantly. In contrast, CREAM only requires storing and evaluating the latest two checkpoints, which is more efficient compared with the standard ensemble methods.

Finally, we would like to put more theoretical remarks on the regularization which is also discussed in Section 3.3 in Fisch et al. (2024). In detail, Fisch et al. (2024) considers the KL divergence between $\pi$ and $\pi_{\text{ref}}$ as

$$\text{KL}(\pi_{\text{ref}} \parallel \pi) \propto \mathbb{E}_x \left[ \int \pi_{\text{ref}}(y \mid x) \log \pi(y \mid x) \, \mathrm{d}y \right] = \mathbb{E}_{x,y \sim \pi_{\text{ref}}} \log \pi(y \mid x), \qquad (1)$$

which is the same as conducting SFT with respect to $\pi_{\text{ref}}$. Ours considers the KL divergence for the preference model, denoted by

$$\text{KL}(u \parallel P_t(y_1 \succ y_2)) \propto \mathbb{E}_{x,y_1,y_2 \sim \pi_{t-1}} \log P_t(y_1 \succ y_2) = \mathbb{E}_{x,y_1,y_2 \sim \pi_{t-1}} \log \sigma \left( \Delta_t^R(y_1, y_2) \right) \quad (2)$$

with $\Delta_t^R(y_1, y_2) = \log \frac{\pi_\theta(y_1 \mid x_t)}{\pi_{\text{ref}}(y_1 \mid x_t)} - \log \frac{\pi_\theta(y_2 \mid x_t)}{\pi_{\text{ref}}(y_2 \mid x_t)}$.

One can find the fundamental differences between (1) and (2):

1. The regularization of (1) is to control the behavior of $\pi$ similar to the reference policy $\pi_{\text{ref}}$. It is more suitable for one iteration of DPO. In the SRLM where the LMs are expected to iteratively improve their performance by themselves, intuitively we should hypothesize that $\pi_t$ becomes more and more distant from $\pi_{\text{ref}}$. As a result, using (1) in SRLM would be too conservative for the LM to improve itself.

2. The regularization of (2) is to prevent the preference model from being overconfident on preference pairs from the last iteration model. Obviously, it fits the SRLM better by letting the SRLM improve from the correct self-labeled preference pairs but also not be too conservative on the previous iteration or the baseline model.

To summarize, we believe CREAM and Fisch et al. (2024) fall into different categories: in SRLM and CREAM, we are more focused on regularizing the preference pairs, while performing model ensemble would be challenging since the model is updating and depends on each other (iteration). We also acknowledge that in regular DPO or with different reward models, the ensemble methods or the regularization methods are feasible and useful.

## C.5 CREAM WITH DATA CONSISTENCY

For CREAM, The introduced (in)consistency serves as the uncertainty quantification for the model to prevent the model itself being overconfident due to the stochastic training noise or incorrectly labeled prefernece data. Such a uncertainty, usually referred to as epistemic uncertainty, is a property for each model. We further add two variants to explore treating the consistency as a property of the data instead of the model: (1) **Threshold = x**: This variant uses the normal DPO for samples (confident dataset) whose consistency rate $C > x$, and uses ours C*DPO + (1-C)*RDPO to regularize the training for other samples (unconfident dataset). (2) **CREAM w/ Dynamic Consistency**: Instead of original CREAM using the average consistency rate across the dataset, this method uses dynamic consistency rate for each sample, using each sample's own consistency rate to regularize the training.

Table 9: Results for treating the consistency as a property of the data instead of the model using CREAM on Llama-3. The best performance among methods in each iteration is highlighted in **bold**.

| Method | Iteration | Arc-Easy | Arc-Challenge | OpenBookQA | SIQA | GSM8K | Average |
|--------|-----------|----------|---------------|------------|------|-------|---------|
| SRLM (Threshold = 0.0) | M2 | 87.79 | 80.38 | 87.80 | **70.95** | 78.01 | 80.99 |
| Threshold = 0.1 | M2 | 88.13 | 80.89 | 88.20 | 70.52 | 78.70 | 81.29 |
| Threshold = 0.3 | M2 | 88.38 | 80.29 | **89.20** | 68.99 | 79.83 | 81.34 |
| Threshold = 0.5 | M2 | **89.10** | 80.89 | 87.20 | 69.70 | 80.52 | 81.48 |
| Threshold = 0.7 | M2 | 88.72 | 80.20 | 88.00 | 69.24 | 80.82 | 81.4 |
| Threshold = 0.9 | M2 | 88.55 | **81.48** | 88.40 | 70.78 | 79.45 | **81.73** |
| CREAM (Threshold = 1.0) | M2 | 88.89 | 80.89 | 88.00 | 69.79 | **81.04** | 81.72 |
| CREAM + dynamic | M2 | 88.13 | 80.80 | 88.00 | 69.50 | 79.38 | 81.16 |
| SRLM (Threshold = 0.0) | M3 | 87.17 | 81.23 | 87.30 | 70.37 | 77.48 | 80.71 |
| Threshold = 0.1 | M3 | 88.38 | 80.29 | 87.20 | 70.98 | 79.83 | 81.34 |
| Threshold = 0.3 | M3 | 88.64 | 80.38 | 88.00 | 71.34 | 80.52 | 81.78 |
| Threshold = 0.5 | M3 | 89.10 | 81.48 | 89.80 | 71.08 | 79.91 | 82.27 |
| Threshold = 0.7 | M3 | 89.48 | 81.31 | 88.60 | 71.55 | 80.67 | 82.32 |
| Threshold = 0.9 | M3 | 89.27 | 83.02 | **90.80** | **72.31** | 81.50 | **83.38** |
| CREAM (Threshold = 1.0) | M3 | **89.52** | **83.36** | 90.20 | 72.06 | **81.73** | 83.37 |
| CREAM + dynamic | M3 | 88.85 | 81.57 | 89.20 | 71.65 | 79.91 | 82.24 |

Table 10: Results of SRLM and CREAM with Llama-2-13B.

| Method | Arc-Easy | Arc-Challenge | OpenBookQA | SIQA | GSM8K | Average |
|--------|----------|---------------|------------|------|-------|---------|
| M0 | 67.47 | 56.31 | 67.00 | 47.54 | 35.03 | 54.67 |
| M1 | 68.27 | 57.42 | 67.40 | 47.85 | 36.09 | 55.41 |
| SRLM M2 | 69.61 | 57.00 | 64.00 | 52.10 | 31.69 | 54.88 |
| SRLM M3 | 62.08 | 53.67 | 61.20 | 48.93 | 20.77 | 49.33 |
| CREAM M2 | 69.19 | 57.17 | 69.60 | 48.57 | 36.01 | 56.11 |
| CREAM M3 | 68.56 | 59.04 | 73.20 | 49.90 | 36.62 | 57.46 |

As shown in Table 9, we observe that as the threshold increases beyond a certain value (including more samples for regularization), the performance gains converge. Actually, CREAM is compatible with using a threshold to fine-grainedly select the data to regularize, however, this inevitably introduces an additional hyperparameter. Besides, CREAM already achieves sufficiently good performance, without this added complexity. Compared to the "CREAM w/ Dynamic Consistency" variant, our method, which utilizes the average consistency rate, significantly reduces the variance in estimating dataset uncertainty, resulting in improved performance.

## C.6 APPLICABILITY OF CREAM

**Applicability to Larger Models.** Due to the limited computational resources, our experiments are mainly conducted on 7B-level LLMs such as Llama-2 and Llama-3. However, we believe this is still meaningful to democratize the self-rewarding paradigm to LLMs of community affordable size, such as 7B models. Advanced and larger LLMs (e.g., ChatGPT) are not always accessible, especially in specific application scenarios like those involving medical or privacy-sensitive data. It's important to note that our method is theoretically applicable to LLMs of any size (not limited to 7B). To validate our method with larger LLMs, we test the CREAM on Llama-2-13B (Touvron et al., 2023) model without any hyperparameter tunings. The results in Table 10 confirm that our method remains effective for the 13B LLM, demonstrating the generalizability of our approach across different model sizes.

**Applicability to unaligned Models.** First, our method CREAM does require the model to have some initial alignment capability, as the adopted DPO rewarding relies on the model being aligned, otherwise the rewards would not be meaningful (Rafailov et al., 2023). Note that in the self-rewarding scenario, the original SRLM (Yuan et al., 2024) also requires the model to be "post-trained" version, where they adopt Llama-2-70B-Chat (Touvron et al., 2023) as the initial model (M0). We conduct additional experiments to test the effectiveness of CREAM with LLMs that are not post-trained. Specifically, we use Llama-3-8B-NO-Chat-Version (Dubey et al., 2024) as the base model (M0). The results in Table 11, reveal that: (1) M0 performs poorly due to its lack of instruc-

Table 11: Results of applying SRLM and CREAM to unaligned Llama-3-8B-NO-Chat-Version.

| Method | Arc-Easy | Arc-Challenge | OpenBookQA | SIQA | GSM8K | Average |
|---|---|---|---|---|---|---|
| M0 | 30.64 | 28.50 | 20.80 | 23.95 | 4.02 | 21.58 |
| M1 | 31.23 | 30.21 | 33.40 | 26.51 | 51.25 | 34.52 |
| SRLM M2 | 29.80 | 28.67 | 36.00 | 23.44 | 50.19 | 33.62 |
| SRLM M3 | 31.82 | 30.03 | 39.00 | 28.40 | 48.52 | 35.55 |
| CREAM M2 | 31.61 | 28.75 | 35.60 | 28.66 | 53.00 | 35.52 |
| CREAM M3 | 39.18 | 35.07 | 49.00 | 33.62 | 56.48 | 42.67 |

Table 12: Results for P-CREAM and Distilled DPO (Fisch et al., 2024).

| Method | Iteration | Arc-Easy | Arc-Challenge | OpenBookQA | SIQA | GSM8K | Average |
|---|---|---|---|---|---|---|---|
| SRLM | M2 | 87.79 | 80.38 | 87.80 | 70.95 | 78.01 | 80.99 |
| P-SRLM | M2 | 84.64 | 76.79 | 80.40 | 67.81 | 78.47 | 77.62 |
| Distilled DPO | M2 | 87.20 | 80.12 | 85.40 | 69.65 | 79.76 | 80.43 |
| P-CREAM | M2 | 87.67 | 78.58 | 86.40 | 68.47 | 79.76 | 80.18 |
| CREAM | M2 | 88.89 | 80.89 | 88.00 | 69.79 | 81.04 | 81.72 |
| SRLM | M3 | 87.17 | 81.23 | 87.30 | 70.37 | 77.48 | 80.71 |
| P-SRLM | M3 | 83.75 | 76.28 | 80.20 | 66.63 | 78.99 | 77.17 |
| Distilled DPO | M3 | 86.15 | 79.01 | 85.00 | 68.68 | 78.62 | 79.49 |
| P-CREAM | M3 | 86.49 | 78.33 | 87.40 | 69.70 | 81.05 | 80.59 |
| CREAM | M3 | 89.52 | 83.36 | 90.20 | 72.06 | 81.73 | 83.37 |

tion following ability. However, after training on a small amount of seed SFT data, M1 demonstrates improved performance, especially on the reasoning task GSM8K. (2) Both SRLM and CREAM can effectively boost the performance, while CREAM often provides greater gains. (3) We find that the performance gains of CREAM in the 3rd iteration (M2 → M3) exceed those in the 2nd iteration (M1 → M2). This may be because M2 has better alignment than M1, and the effectiveness of our method relies on the model's alignment capability. Based on these findings, we conclude that SRLM-like methods can still be applied to pre-trained models, provided they have undergone slight alignment (e.g., through SFT training).

## C.7 PROMPT-REWARDING FOR REGULARIZATION

In this section, we discuss the usage of prompt-rewarding with regularization, and compare its effectiveness of the DPO rewarding. We mainly consider two methods: (1) P-CREAM: CREAM with prompt-rewarding method. (2) Distilled DPO (Fisch et al., 2024) which leverages the prompt-rewarding to apply regularization to the DPO training. The results are shown in Table 12. We can find that (1) prompt-rewarding is not effective as DPO rewarding for both SRLM and CREAM. (2) The degradation of Distilled DPO in M3 iteration indicates its inapplicability for continuous improvements, compared to our method CREAM.

Table 13: Results for different self-rewarding methods using Llama-3.

| Method / Dataset | M1 SFT | M2 SRLM | M2 P-SRLM | M2 CREAM | M3 SRLM | M3 P-SRLM | M3 CREAM |
|---|---|---|---|---|---|---|---|
| Arc-Easy | 86.78 | 87.79 ↑ | 84.64 ↓ | **88.89** ↑ | 87.17 ↓ | 83.75 ↓ | **89.52** ↑ |
| Arc-Challenge | 80.14 | 80.38 ↑ | 76.79 ↓ | **80.89** ↑ | 81.23 ↑ | 76.28 ↓ | **83.36** ↑ |
| OpenBookQA | 86.40 | 87.80 ↑ | 80.40 ↓ | **88.00** ↑ | 87.30 ↓ | 80.20 ↓ | **90.20** ↑ |
| SIQA | 69.50 | **70.95** ↑ | 67.81 ↓ | 69.79 ↑ | 70.37 ↓ | 66.63 ↓ | **72.06** ↑ |
| GSM8K | 78.39 | 78.01 ↓ | 78.47 ↑ | **81.04** ↑ | 77.48 ↓ | 78.99 ↑ | **81.73** ↑ |

