# OpenReview forum: "CREAM: Consistency Regularized Self-Rewarding Language Models"
_ICLR.cc/2025/Conference — ICLR 2025 Poster_

### Official Review · Reviewer_LeV2 · 2024-11-01

**Soundness:** 3
**Presentation:** 3
**Contribution:** 3
**Rating:** 6
**Confidence:** 3

**Summary:**

This paper proposes a form of regularization for self-rewarding language models, which involves creating a loss that is a weighted combination of the LM's implied preferences and the reverse preferences. The weighting is determined by the rank correlation between implied rewards across adjacent training steps: if these rewards are perfectly rank-correlated then the reverse preferences are not included; to the extent that they are uncorrelated, the self reward model is deemed less trustworthy and so the reverse preferences are given more weight. The specific form of the regularizer is justified by theoretical connections to entropic regularization. The method yields improvements over the unregularized SRLM approach of Yuan et al (2024) on benchmark datasets and in head-to-head win rate (e.g., 34/21 win-loss ratio). Evaluations focus at the 7B scale.

**Strengths:**

- While the proposed approach seems somewhat heuristic (e.g., training on both (y, y') and (y', y)), the steps of the algorithm are justified by theoretical support.
- The "oracle" component of the evaluation helps to clarify the headroom for better reward modeling on these tasks.
- The ablations help clarify the role of the specific form of rank correlation and "prompt rewarding" vs the implicit DPO RM

**Weaknesses:**

It's not clear to me why one would want to use self-rewarding LMs at the 7B scale, as it would not be too difficult to get preference annotations from a larger model and train on them instead.

The algorithm appears to treat consistency as a property of the model \theta_t, but I wonder whether it would be better to treat it as a property of the *data*. That is, for any model, there will be some examples for which preference can be assessed confidently, and others for which it cannot. It would be ideal to use the full DPO loss for the first type of examples, and regularize strongly on the second type. Instead, the proposed approach seems to set the regularization parameter based on some property of the optimization path: as long as the LM has not changed its rankings from time t-1, it will use the standard DPO objective. (Thanks for the additional experiments on this.)

I would like to see more analysis of the sum of DPO losses. In particular, what is the effect to the policy of applying the update in step 14 of the algorithm when C=1/2? My guess is that the probability for both y+ and y- decreases, but it would be great to have some formal analysis because this situation could be relevant, e.g., at the beginning of training.

**Questions:**

The method seems to be applied to LMs that have already been post-trained (LLAMA 2 & 3), which is perhaps why the impact is less than in Yuan et al. Would it be possible to apply the method to a model immediately after pretraining or instruction tuning?

---

> ### Author Response · Authors · 2024-11-21
> **Response to Reviewer LeV2 (1 / 4)**
>
> Thank you for your constructive comments! We respond to your questions below and would appreciate it if you could let us know if our response helps address your concerns.
>
>
> > Weakness 1: It's not clear to me why one would want to use self-rewarding LMs at the 7B scale, as it would not be too difficult to get preference annotations from a larger model and train on them instead.
>
> **Response 1**:
>
> * First, we believe it's meaningful to democratize the self-rewarding paradigm to LLMs of community affordable size, such as 7B models. Advanced and larger LLMs (e.g., ChatGPT) are not always accessible, especially in specific application scenarios like those involving medical or privacy-sensitive data.
> * We want to emphasize that our method is theoretically applicable to LLMs of any size (not limited to 7B). However, we present empirical results with 7B LLMs due to limited computational resources.
> * To validate our method with larger LLMs, we rented h100 gpus to test the CREAM on Llama-2-13B model without any hyperparameter tuning. The results confirm that our method remains effective for the 13B LLM.
>
> | Method   | Arc-Easy | Arc-Challenge | OpenBookQA | SIQA  | GSM8K | Average |
> | -------- | -------- | ------------- | ---------- | ----- | ----- | ------- |
> | M0       | 67.47    | 56.31         | 67.00      | 47.54 | 35.03 | 54.67   |
> | M1       | 68.27    | 57.42         | 67.40      | 47.85 | 36.09 | 55.41   |
> | SRLM M2  | 69.61    | 57.00         | 64.00      | 52.10 | 31.69 | 54.88   |
> | SRLM M3  | 62.08    | 53.67         | 61.20      | 48.93 | 20.77 | 49.33   |
> | CREAM M2 | 69.19    | 57.17         | 69.60      | 48.57 | 36.01 | 56.11   |
> | CREAM M3 | 68.56    | 59.04         | 73.20      | 49.90 | 36.62 | 57.46   |

---

> ### Author Response · Authors · 2024-11-21
> **Response to Reviewer LeV2 (2 / 4)**
>
> > Weakness 2: The algorithm appears to treat consistency as a property of the model \theta_t, but I wonder whether it would be better to treat it as a property of the data. That is, for any model, there will be some examples for which preference can be assessed confidently, and others for which it cannot. It would be ideal to use the full DPO loss for the first type of examples, and regularize strongly on the second type. Instead, the proposed approach seems to set the regularization parameter based on some property of the optimization path: as long as the LM has not changed its rankings from time t-1, it will use the standard DPO objective.
>
> **Response 2**:  Thank you for your suggestion. The introduced (in)consistency serves as the uncertainty quantification for the model to prevent the model itself being overconfident due to the stochastic training noise or incorrectly labeled prefernece data. Such a uncertainty, usually referred to as epistemic uncertainty, is a property for each model.
>
> We further add two variants to explore treating the consistency as a poperty of the data instead of the model:
> * **Threshold = x**: This variant uses the normal DPO for samples (confident dataset) whose consistency rate $C > x$, and uses ours C*DPO + (1-C)*RDPO to regularize the training for other samples (unconfident dataset).
> * **CREAM w/ Dynamic Consistency**: Instead of original CREAM using the average consistency rate across the dataset, this method uses dynamic consistency rate for each sample, using each sample's own consistency rate to regularize the training.
>
> The results are shown as below.
>
> * We observe that as the threshold increases beyond a certain value (including more samples for regularization), the performance gains converge. Actually, CREAM is compatible with using a threshold to fine-grainedly select the data to regularize, however, this inevitably introduces an additional hyperparameter. Besides, CREAM already achieves sufficiently good performance, without this added complexity.
> * Compared to the "CREAM w/ Dynamic Consistency" variant, our method, which utilizes the average consistency rate, significantly reduces the variance in estimating dataset uncertainty, resulting in improved performance.
>
> | Method                  | Iteration | Arc-Easy | Arc-Challenge | OpenBookQA | SIQA  | GSM8K | Average |
> | ----------------------- | --------- | -------- | ------------- | ---------- | ----- | ----- | ------- |
> | Threshold = 0.0 (SRLM)  | M2        | 87.79    | 80.38         | 87.80      | 70.95 | 78.01 | 80.99   |
> | Threshold = 0.1         | M2        | 88.13    | 80.89         | 88.20      | 70.52 | 78.70 | 81.29   |
> | Threshold = 0.3         | M2        | 88.38    | 80.29         | 89.20      | 68.99 | 79.83 | 81.34   |
> | Threshold = 0.5         | M2        | 89.10    | 80.89         | 87.20      | 69.70 | 80.52 | 81.48   |
> | Threshold = 0.7         | M2        | 88.72    | 80.20         | 88.00      | 69.24 | 80.82 | 81.4    |
> | Threshold = 0.9         | M2        | 88.55    | 81.48         | 88.40      | 70.78 | 79.45 | 81.73   |
> | CREAM (Threshold = 1.0) | M2        | 88.89    | 80.89         | 88.00      | 69.79 | 81.04 | 81.72   |
> | CREAM w/ Dynamic Consistency         | M2        | 88.13    | 80.80         | 88.00      | 69.50 | 79.38 | 81.16   |
> | SRLM (Threshold = 0.0)  | M3        | 87.17    | 81.23         | 87.30      | 70.37 | 77.48 | 80.71   |
> | Threshold = 0.1         | M3        | 88.38    | 80.29         | 87.20      | 70.98 | 79.83 | 81.34   |
> | Threshold = 0.3         | M3        | 88.64    | 80.38         | 88.00      | 71.34 | 80.52 | 81.78   |
> | Threshold = 0.5         | M3        | 89.10    | 81.48         | 89.80      | 71.08 | 79.91 | 82.27   |
> | Threshold = 0.7         | M3        | 89.48    | 81.31         | 88.60      | 71.55 | 80.67 | 82.32   |
> | Threshold = 0.9         | M3        | 89.27    | 83.02         | 90.80      | 72.31 | 81.50 | 83.38   |
> | Threshold = 1.0 (CREAM) | M3        | 89.52    | 83.36         | 90.20      | 72.06 | 81.73 | 83.37   |
> | CREAM w/ Dynamic Consistency         | M3        | 88.85    | 81.57         | 89.20      | 71.65 | 79.91 | 82.24   |

---

> ### Author Response · Authors · 2024-11-21
> **Response to Reviewer LeV2 (3 / 4)**
>
> > Weakness 3: I would like to see more analysis of the sum of DPO losses. In particular, what is the effect to the policy of applying the update in step 14 of the algorithm when C=1/2? My guess is that the probability for both y+ and y- decreases, but it would be great to have some formal analysis because this situation could be relevant, e.g., at the beginning of training.
>
>
> **Response 3**: Thank you for your question! Here we take the gradient analysis for the sum of the DPO losses.
>
> * Recall that CREAM uses the $L = C * D + (1-C) * RD$ for training, where $C$ is the consistency rate, $D$ and $RD$ are DPO loss and ReverseDPO loss, respectively.
> * We can simplify the DPO loss by substituting the rewarding gap $[\log [\pi_\theta(y+) / \pi_\text{ref}(y+)] - \log [\pi_\theta(y-) / \pi_\text{ref}(y-)]]$ as $x_\theta$, then, the $D$ loss is written as $\log \sigma (x_\theta)$, and $RD$ loss is written as $\log \sigma (-x_\theta)$.
> * Recall that the sigmoid function $\sigma$ has the poperty that $\sigma(x) = 1 - \sigma(-x)$. Then, $RD$ can be converted to $\log (1-\sigma(x_\theta))$. Our regularized loss $L$ is $C \log \sigma (x_\theta) + (1-C)\log (1-\sigma(x_\theta))$.
> * The gradient of $L$ with respect to model params $\theta$ is $\frac{\partial L}{\partial \theta} = (C-\sigma(x_\theta)) \frac{\partial x_\theta}{\partial \theta}$.
> * When $C = 1/2$, it is pushing the model to learn $\sigma (x_\theta) = 1/2$, i.e., encouraging the model to have $[\pi_\theta(y+) / \pi_\text{ref}(y+)] = [\pi_\theta(y-) / \pi_\text{ref}(y-)]$. And so far there is no theoretical evidence that the likelihoods of both y+ and y- will decrease. However, based on empirical results of existing works[1,2,3], the $\frac{\partial x_\theta}{\partial \theta}$ part will decrease both likelihood of y+ and y- but increase their gap. Thus, the combined optimization will potentially lead to:
>     * decrease both likelihood of y+ and y-
>     * increase the gap between y+ and y- but stay close to $[\pi_\theta(y+) / \pi_\text{ref}(y+)] = [\pi_\theta(y-) / \pi_\text{ref}(y-)]$
>
>
> Additionally, we supply experiment results of CREAM with $C=1/2$ as follows. $C=1/2$ is generally better than normal SRLM, which confirms the effectiveness of the proposed regularization.
>
> | Method      | Iteration | Arc-Easy | Arc-Challenge | OpenBookQA | SIQA  | GSM8K | Average |
> | ----------- | --------- | -------- | ------------- | ---------- | ----- | ----- | ------- |
> | SRLM        | M2        | 87.79    | 80.38         | 87.80      | 70.95 | 78.01 | 80.99   |
> | CREAM+C=0.5 | M2        | 88.17    | 80.89         | 85.00      | 70.88 | 79.15 | 80.82   |
> | CREAM       | M2        | 88.89    | 80.89         | 88.00      | 69.79 | 81.04 | 81.72   |
> | SRLM        | M3        | 87.17    | 81.23         | 87.30      | 70.37 | 77.48 | 80.71   |
> | CREAM+C=0.5 | M3        | 87.29    | 80.80         | 85.00      | 67.66 | 80.06 | 80.16   |
> | CREAM       | M3        | 89.52    | 83.36         | 90.20      | 72.06 | 81.73 | 83.37   |
>
> [1] Rafailov, Rafael, et al. "Direct preference optimization: Your language model is secretly a reward model." Advances in Neural Information Processing Systems 36 (2024).
>
> [2] Azar, Mohammad Gheshlaghi, et al. "A general theoretical paradigm to understand learning from human preferences." International Conference on Artificial Intelligence and Statistics. PMLR, 2024.
>
> [3] Park, Ryan, et al. "Disentangling length from quality in direct preference optimization." arXiv preprint arXiv:2403.19159 (2024).

---

> ### Author Response · Authors · 2024-11-21
> **Response to Reviewer LeV2 (4 / 4)**
>
> > Question 1: The method seems to be applied to LMs that have already been post-trained (LLAMA 2 & 3), which is perhaps why the impact is less than in Yuan et al. Would it be possible to apply the method to a model immediately after pretraining or instruction tuning?
>
> **Response 4**:
>
> * Our method does require the model to have some initial alignment capability, as the adopted DPO rewarding relies on the model being aligned, otherwise the rewards would not be meaningful. The relatively weaker results of Llama-2 compared to Llama-3 can support this point.
> * Note that in the self-rewarding scenario, the original paper SRLM[4] also requires the model to be "post-trained" version, where they adopt Llama-2-70B-Chat as the initial model (M0).
> * We conduct additional experiments to test the effectiveness of CREAM with LLMs that are not post-trained. Specifically, we use Llama-3-8B-NO-Chat-Version as the base model. The results, shown below, reveal that:
>     *  M0 performs poorly due to its lack of instruction following ability. However, after training on a small amount of seed SFT data, M1 demonstrates improved performance, especially on the reasoning task GSM8K.
>     *  Both SRLM and CREAM can effectively boost the performance, while CREAM often provides greater gains.
>     *  We find that the performance gains of CREAM in the 3rd iteration (M2 -> M3) exceed those in the 2nd iteration (M1 -> M2). This may be because M2 has better alignment than M1, and the effectiveness of our method relies on the model's alignment capability.
>
> Based on these experimental results, we conclude that SRLM-like methods can still be applied to pre-trained models, provided they have undergone slight alignment (e.g., through SFT training).
>
>
> | Method   | Arc-Easy | Arc-Challenge | OpenBookQA | SIQA  | GSM8K | Average |
> | -------- | -------- | ------------- | ---------- | ----- | ----- | ------- |
> | M0       | 30.64    | 28.50         | 20.80      | 23.95 | 4.02  | 21.58   |
> | M1       | 31.23    | 30.21         | 33.40      | 26.51 | 51.25 | 34.52   |
> | SRLM M2  | 29.80    | 28.67         | 36.00      | 23.44 | 50.19 | 33.62   |
> | SRLM M3  | 31.82    | 30.03         | 39.00      | 28.40 | 48.52 | 35.55   |
> | CREAM M2 | 31.61    | 28.75         | 35.60      | 28.66 | 53.00 | 35.52   |
> | CREAM M3 | 39.18    | 35.07         | 49.00      | 33.62 | 56.48 | 42.67   |
>
> [4] Yuan, Weizhe, et al. "Self-Rewarding Language Models." Forty-first International Conference on Machine Learning.

---

> > ### Comment · Reviewer_LeV2 · 2024-11-23
> > **Thanks for the additional experiments**
> >
> > Thanks for these experiments, particularly the thresholded CREAM results -- very helpful! I will increase my score.
> >
> > Just one note regarding this point:
> >
> > > It's not clear to me why one would want to use self-rewarding LMs at the 7B scale, as it would not be too difficult to get preference annotations from a larger model and train on them instead.
> >
> > I'm not suggesting the paper requires testing self-rewarding LMs at larger scales (although I appreciate the additional experiment) but that I wonder whether it wouldn't just be better to apply a more traditional RLAIF setup if the policy model is this small.

---

> > > ### Author Response · Authors · 2024-11-24
> > > **Thank you for your recognition!**
> > >
> > > Thank you very much for your time and the decision to raise the score! We are happy to see that our responses and experiments are helpful to address your concerns.
> > >
> > > We would like to further clarify the motivation of applying SRLM with 7B level small LLMs:
> > > > why not using RLAIF (with external reward models) with small LLMs
> > >
> > > - First, we think it is meaningful to democratize the self-rewarding ability to 7B level small LLMs. There are **many realistic scenarios requiring small LLMs while external reward models are not available**, while SRLMs can come to help:
> > >     * Low-resource domain that have no existing available reward models or existing general reward models have distribution-shift problem. For example, GPT-4 may not handle well on extremely low-resource languages. In such case, SRLM can be a good choice for self-improvements.
> > >     * Privacy senstive and edge computing scenarios that the deployed models often possess user's personal privacy data or need to compute offline, thus these models cannot utilize external reward models for continuous improvements. And training an internal reward model can also be hard due to lack of annotations.
> > >
> > > - Second, our "Oracle" variant which uses an external reward model and ground truth labels for rewarding in Table 1 is actually a strong baseline of RLAIF, which easily beats SRLMs. However, our CREAM in M3 iteration with Llama-3 shows competitive performance with "Oracle", without relying on any external models or labels, which demonstrates the potential of SRLMs or self-improvements.
> > > - Finally, we want to emphasize that the proposed method and regularization are theoretically applicable to LLMs with any size.
> > >
> > > Thank you again for your time and constructive reviews!

---

### Official Review · Reviewer_AuNi · 2024-11-01

**Soundness:** 2
**Presentation:** 2
**Contribution:** 2
**Rating:** 6
**Confidence:** 3

**Summary:**

This paper proposes a modified DPO objective that considers multiple reward judgements, and penalizes disagreement across multiple reward models. The new objective is studied in the context of self-rewarding LMs, an iterative process based on DPO.

(Note: score increased from 3 to 6 based on the discussion)

**Strengths:**

* The idea of considering diversity across an ensemble of RMs to prevent reward hacking is sound and well motivated.
* Consideration of such a signal is supported by the experimental results.

**Weaknesses:**

* The biggest weakness of this paper is that it is not contextualized well with prior work on reward hacking and using ensembles to mitigate this. Given that proposing variants of DPO and related algorithms is increasingly crowded, this is especially important. Beyond discussing and comparing with this prior work, it is also important to understand how pessimism-based approaches compare, both theoretically and empirically, with the proposed methods based on agreement metrics. Similarly to this paper, there is a growing body of work that aims to mitigate reward hacking through reward model ensembles (e.g. https://arxiv.org/abs/2310.02743, https://arxiv.org/abs/2312.09244, etc.). These papers provide analysis showing various correlation metrics during RLHF. “Disagreement” between reward models can be penalized by considering pessimistic aggregations of rewards, e.g. by taking the min or some percentile (<50) over the set of RM rewards. Such concepts have also previously been integrated into DPO (e.g. https://arxiv.org/abs/2405.19316). It is not clear the relation between the proposed approach of weighting by RM agreement to this prior work which aggregates RM scores pessimistically, or whether there is any advantage.
* The motivation for introducing the framework in 3.1 was not clear to me. How is it different from prior work, e.g. Yuan et al., 2024? Why is it necessary to introduce this new framework to evaluate the key contribution of the paper, i.e. the impact of the consistency reward? This made the experiments more difficult to analyze, because it was unclear whether the only change between "SRLM" and "CREAM" was the consistency reward or whether other factors contributed to the difference.
* The overly formal exposition of the proposed method makes the intuition difficult to follow. For example, the definition of the regularization loss, introduced in equation 3.5, does not include any intuition for why this term was chosen.

**Questions:**

See weaknesses for questions.

There are many grammar mistakes that would be good to resolve, e.g.:
* Abstract - “the accurate of the” -> “the accuracy of the”
* Intro - “we formulates” -> “we formulate”
* Many places - “the rewarding consistency” -> “the consistency reward”?
* Section 3.2 - “prevents the model… from overconfident”

Other nits:
* Notation - You define `[k]` as a set, but the next sentence says “Sets are denoted by calligraphic uppercase letters”. Maybe say “Other sets are…” to avoid a contradiction?
* Equation 3.4 - My understanding is that typically SFT is performed first, and then DPO, rather than optimizing both objectives jointly, as this equation implies.
* The RewardBench citation is missing a year.

---

> ### Author Response · Authors · 2024-11-21
> **Response to Reviewer AuNi (1 / 3)**
>
> Thank you for your time and efforts reviewing our paper. We sincerely value your insightful suggestions and feedback. We would like to address your concerns by point-to-point response as follows.
>
>
> > Weakness 1: The biggest weakness of this paper is that it is not contextualized well with prior work on reward hacking and using ensembles to mitigate this. Given that proposing variants of DPO and related algorithms is increasingly crowded, this is especially important. Beyond discussing and comparing with this prior work, it is also important to understand how pessimism-based approaches compare, both theoretically and empirically, with the proposed methods based on agreement metrics. Similarly to this paper, there is a growing body of work that aims to mitigate reward hacking through reward model ensembles (e.g. https://arxiv.org/abs/2310.02743, https://arxiv.org/abs/2312.09244, etc.). These papers provide analysis showing various correlation metrics during RLHF. “Disagreement” between reward models can be penalized by considering pessimistic aggregations ......
>
> **Response 1**: Thank you for pointing out those related works. We will add a paragraph in Sec. 2 Related Work in the updated version, discussing the relationship between these works and CREAM. In particular, the major difference is that both reward hacking and ensemble methods are dealing with the reward model. In this case, these works add the conservative value estimation for the estimated reward. However, in CREAM, we are only collecting the preference data using LLM's self-assessment and **do not use the exact value of the reward**. In this context, adding the conservative value function to the reward might not be effective because **we are comparing the estimated rewards for preference data instead of maximizing the estimated reward function**.
>
> To tackle this problem, CREAM employs regularization to mitigate the over-estimation of the reward. We would like to point out that in the offline RL literature[1,2,3], both regularization and conservative value estimation (such as taking the minimum value over a set of estimation) are well-known methods for implementing the 'pessimism' in offline RL. Specifically, CREAM considers the regularization as the KL divergence between the preference model $P_{\theta}$ and the uniform Bernouli distribution so that the regularization would prevent the LLM from being overconfidence to one response $y$ compared with the other response $y'$. We would also like to highlight that **this regularization is extremely important in SRLMs, where both $y$ and $y'$ are generated by the language model itself during the training process**.
>
> [1] Kumar, Aviral, et al. "Conservative q-learning for offline reinforcement learning." Advances in Neural Information Processing Systems 33 (2020): 1179-1191.
>
> [2] Farahmand, Amir, et al. "Regularized policy iteration." Advances in Neural Information Processing Systems 21 (2008).
>
> [3] Schulman, John. "Trust Region Policy Optimization." arXiv preprint arXiv:1502.05477 (2015).
>
> In addition to this dissusion, we also include an ablation study on using the aforementioned **conservative value estimation methods (Ensemble-Worst) and Ensemble-Mean** in SRLMs. Specifically, ensemble method trains 3 different $\pi_\theta$ models with 3 different learning rates [7e-7, 1e-6, 3e-6] to serve as the ensemble models in each iteration, since no external models can be involved in the self-rewarding scenario. Then, these three models would independently reward and rank the responses:
> * **Ensemble-Worst** selects the minimum reward for ranking the response.
> * **Ensemble-Mean** selects the average reward for ranking the response.
>
> The results below suggest that CREAM has advantage against Ensemble methods in the alignment performance a cross iterations. Additionally, compared with the ensemble model which requires a batch of reward models, CREAM only requires the model from the last one iteration and thus would be more efficient than the ensemble methods.
>
> | Method         | Iteration | Arc-Easy | Arc-Challenge | OpenBookQA | SIQA  | GSM8K | Average |
> | -------------- | --------- | -------- | ------------- | ---------- | ----- | ----- | ------- |
> | SRLM           | M2        | 87.79    | 80.38         | 87.80      | 70.95 | 78.01 | 80.99   |
> | Ensemble-Worst | M2        | 88.00    | 79.78         | 87.00      | 69.96 | 78.54 | 80.66   |
> | Ensemble-Mean  | M2        | 88.47    | 80.80         | 88.00      | 69.50 | 80.67 | 81.49   |
> | CREAM          | M2        | 88.89    | 80.89         | 88.00      | 69.79 | 81.04 | 81.72   |
> | SRLM           | M3        | 87.17    | 81.23         | 87.30      | 70.37 | 77.48 | 80.71   |
> | Ensemble-Worst | M3        | 87.75    | 80.89         | 86.80      | 70.78 | 78.70 | 80.98   |
> | Ensemble-Mean  | M3        | 88.85    | 80.89         | 87.00      | 69.96 | 79.98 | 81.34   |
> | CREAM          | M3        | 89.52    | 83.36         | 90.20      | 72.06 | 81.73 | 83.37   |

---

> > ### Comment · Reviewer_AuNi · 2024-11-22
> >
> > Thank you for your response and new results. I will take your response and new results into consideration, but do have one follow up question first.
> >
> > > However, in CREAM, we are only collecting the preference data using LLM's self-assessment and do not use the exact value of the reward. In this context, adding the conservative value function to the reward might not be effective because we are comparing the estimated rewards for preference data instead of maximizing the estimated reward function.
> >
> > I agree that simply using conservative reward estimates in the context of determining pairwise preferences is probably not the best approach. However, I thought, for example, that the methods of Fisch et al. 2024 (https://arxiv.org/abs/2405.19316) would actually be quite applicable? Using the "Distillation" loss variant of DPO (their eq. 7) based on the rewards assigned by the LLM could presumably help mitigate over-confidence, similarly to the proposed objective. Perhaps the ensemble and/or pessimistic version of their loss could also be applied, e.g. using the LLM at the current and previous time steps to form the members of the ensemble. Would this be similar to CREAM? Baselines such as these were what I was hoping the authors could discuss. I don't expect the authors to run new experiments given the limited time remaining in the discussion period, but some discussion would be helpful.
> >
> > My concern is just that whether the complexity of CREAM is well justified and there are not simpler methods from prior work that would be similarly effective, especially given the quantity of papers in this area already. But I also acknowledge that I may have misunderstood parts of the method or the relevance of some prior work.

---

> > > ### Author Response · Authors · 2024-11-23
> > > **Additional Response to Reviewer AuNi (1 / 2)**
> > >
> > > Thank you for your prompt reply. We appriciate your recognition on simply using conservative reward estimates in the context of determining pairwise preferences is probably not the best approach. We also appreciate pointing out that important related work[1] and we have updated a revision including a detailed discussion on that in both **Sec. Related Work and Appendix C.5**. We are happy to discuss the connection and difference between Fisch et al. 2024[1]. **We are also working on experimental comparison with the persmestic ensemble dpo methods[1], and will try our best to deliver the experimental results.** From a methodological perspective, the following is a detailed discussion.
> > >
> > >
> > > In detail, Fisch et al. 2024[1] suggested to inject the pessimism by ensemble or introducing the KL divergence $\mathrm{KL}(\pi_{\theta_t} \parallel \pi_{\text{ref}})$. However, we would like to highlight that the methods suggested by Fisch et al. 2024 **cannot be directly applied in our task scenario, due to the nature of self-rewarding setting.**
> > >
> > > First, regarding the **reward distillation**, Fisch et al. 2024[1] suggested to distill the (ensembled) reward information into the language models. However, the **knowledge distillation** usually assumes to distill the information from a large model into a smaller model[2,3,4]. This makes perfect sense when the reward is **given** (e.g. from human peference or an oracle reward model or a more advanced model). However, in SRLMs, the reward preference is given by the language model in the last iteration. The model capacity does not change over time so the distillation may not help improve the model. For instance, if we extract $r^*$ from language model $\pi_{\theta_{t-1}}$, the reward distillation (eq. 7 in Fisch et al. 2024) becomes $$\theta_t = \arg\min_{\theta} \mathbb E\left(\log \frac{\pi_{\theta_{t-1}}(y_+ | x)}{\pi_{\theta_{t-1}}(y_- | x)} - \log \frac{\pi_{\theta}(y_+ | x)}{\pi_{\theta}(y_- | x)}\right)^2,$$
> > >
> > > and a trival minimizer is $\theta_t = \theta_{t-1}$ with loss equals to zero. This indicates that if we use the same, single model for reward distillation, the model can hardly be improved.
> > >
> > > Second, we would like to acknowledge that the ensembled model used in Fisch et al. 2024[1] and other works[5,6] can provide the pessimism, especially when we have a batch of reward models. For example, Fisch et al. 2024[1] used 5 reward models in their Pessimistic Ensemble DPO (e-DPO) variant which has the best performance. However, in the **self-rewarding** setting, each reward model are trained by ourself so **obtaining a reward model would be costly**.
> > >
> > > For example, let's assume $\pi_{\text{ref}}$ is the reference model and $\pi_1$ is the model trained by SFT, $\pi_2$ is the 1st round of self-rewarding training and $\pi_3$ is the 2nd round, etc. As you suggested to
> > > > use the LLM at the current and previous time steps to form the members of the ensemble,
> > >
> > > We would see such an ensemble will not work for training $\pi_2$, and for $\pi_3$ will ensemble 2 models, $\pi_4$ will ensemble 3 models. Note that though we can use different data subsets or hyperparams to obtain multiple reward models in the earlier iterations, it can bring additional training costs. It is obvious that this algorithm needs to store all models $\pi_1, \pi_2, \cdots \pi_M$ as the iteration goes up. In the case where more models are required to improve the ensemble performance (e.g. we need to ensemble $5$ seperate $\pi_1$ for Pessimistic Ensemble DPO), this storage and computation comsumption goes even higher. In contrast, CREAM only require to store and evaluate the latest two checkpoints, which would be **more efficient** compared with the standard ensemble methods.

---

> > > ### Author Response · Authors · 2024-11-23
> > > **Additional Response to Reviewer AuNi (2 / 2)**
> > >
> > > Finally, we would like to put more **theoretical remarks on the regularization** which is also discussed in Section 3.3 in Fisch et al. 2024[1]. In detail, Fisch et al. 2024[1] considers the KL divergence between $\pi$ and $\pi_{\text{ref}}$ as
> > > $$\mathrm{KL}(\pi_{\text{ref}} \parallel \pi) \propto \mathbb E_x [ \textstyle{\int \pi_{\text{ref}}(y | x) \log \pi(y | x) \mathrm dy}] = \mathbb E_{x, y \sim \pi_{\text{ref}}} \log \pi(y | x), \qquad \qquad (1)$$
> > > which is the same with conducting SFT w.r.t. $\pi_{\text{ref}}$. Ours considers the KL divergence for the preference model, denoted by
> > > $$\mathrm{KL}(u \parallel P_t(y_1 \succ y_2)) \propto \mathbb E_{x, y_1, y_2 \sim \pi_{t-1}} \log P_t(y_1 \succ y_2) = \mathbb E_{x, y_1, y_2 \sim \pi_{t-1}} \log \sigma\left(\Delta^R_t(y_1, y_2)\right) \quad (2) $$
> > > with $\Delta^R_t(y_1, y_2) = \log \frac{\pi_\theta(y_1 | x_t)}{\pi_{\text{ref}}(y_1 | x_t)} - \log \frac{\pi_\theta(y_2 | x_t)}{\pi_{\text{ref}}(y_2 | x_t)}$.
> > >
> > > One can find the fundamental differences between (1) and (2):
> > > 1. The regularization of (1) is to control the behavior of $\pi$ similar with the reference policy $\pi_{\text{ref}}$. It is more suitable for one iteration of DPO. In the SRLM where the LMs are expected to iterativly improve its performance by itself, intuitively we should hypothesis that $\pi_t$ becomes more and more away from $\pi_{\text{ref}}$. As a result, using (1) in SRLM would be too conservative for the LM to improve itself.
> > > 2. The regularization of (2) is preventing the preference model being overconfidence on **preference pair** from the **last iteration model**. Obviously it fits the SRLM better by letting the SRLM to improve from the correct self-labeled preferences pairs but also not too conservative on the previous iteration or the baseline model.
> > >
> > > To summarize, we believe CREAM and the paper[1] you mentioned falls into different categories: in SRLM and CREAM, we are more **focused on regularizing the preference pair**, while performing the model ensemble would be challenging: since the model is updating and depends on each other (iteration). We also acknowledge that in the regular DPO or with different reward models, the ensemble methods or the regularization presented in the papers you pointed out are feasible and useful.
> > >
> > > ---
> > >
> > > **References**:
> > >
> > > [1] Fisch, Adam, et al. "Robust preference optimization through reward model distillation." arXiv preprint arXiv:2405.19316 (2024).
> > >
> > > [2] Hinton, Geoffrey. "Distilling the Knowledge in a Neural Network." arXiv preprint arXiv:1503.02531 (2015).
> > >
> > > [3] Mirzadeh, Seyed Iman, et al. "Improved knowledge distillation via teacher assistant." Proceedings of the AAAI conference on artificial intelligence. Vol. 34. No. 04. 2020.
> > >
> > > [4] Buciluǎ, Cristian, Rich Caruana, and Alexandru Niculescu-Mizil. "Model compression." Proceedings of the 12th ACM SIGKDD international conference on Knowledge discovery and data mining. 2006.
> > >
> > > [5] Coste, Thomas, et al. "Reward model ensembles help mitigate overoptimization." arXiv preprint arXiv:2310.02743 (2023).
> > >
> > > [6] Eisenstein, Jacob, et al. "Helping or herding? reward model ensembles mitigate but do not eliminate reward hacking." arXiv preprint arXiv:2312.09244 (2023).

---

> ### Author Response · Authors · 2024-11-21
> **Response to Reviewer AuNi (2 / 3)**
>
> > Weakness 2: The motivation for introducing the framework in 3.1 was not clear to me. How is it different from prior work, e.g. Yuan et al., 2024? Why is it necessary to introduce this new framework to evaluate the key contribution of the paper, i.e. the impact of the consistency reward? This made the experiments more difficult to analyze, because it was unclear whether the only change between "SRLM" and "CREAM" was the consistency reward or whether other factors contributed to the difference.
>
> **Response 2**: We would like to highlight our contribution by proposing a universal framework including several existing iterative preference fine-tuning works, which not only includes the work of Yuan et al., 2024 (SRLM)[4], but also covers other iterative eference fine-tuning works[5,6]. This universal framework **enables us to better design some theoretical-grounded methods tackling with the limitations of the existing algorithms**, such as the overconfident in SRLMs. As in CREAM, we propose adding consistency-based regularization as presented in Eq. 3.5 for SRLM. Another contribution of the proposed framework is that **we can convert the regularzation into a easier-to-implement label smoothing DPO method**.
>
> In conclusion, although the difference between the CREAM and SRLM mainly lies in calculating the consistency rate and using it in DPO training stage, the CREAM delivers more theoretical insights on introducing implicit regularization through this consistency rate and offering a universal understanding of the iterative preference-based fine-tuning works.
>
> [4] Yuan, Weizhe, et al. "Self-Rewarding Language Models." Forty-first International Conference on Machine Learning.
>
> [5] Lee, Harrison, et al. "RLAIF: Scaling Reinforcement Learning from Human Feedback with AI Feedback." arXiv e-prints (2023): arXiv-2309.
>
> [6] Wu, Yue, et al. "Self-play preference optimization for language model alignment." arXiv preprint arXiv:2405.00675 (2024).
>
>
> ---
>
> > Weakness 3: The overly formal exposition of the proposed method makes the intuition difficult to follow. For example, the definition of the regularization loss, introduced in equation 3.5, does not include any intuition for why this term was chosen.
>
> **Response 3**: **The intuition behind the regularization loss in Eq. (3.5) is explained as follows**: Consider two response $y$ and $y'$ generated by the same language model using the same input prompt $x$. When the language model is required to self-judge the preference order between $y$ and $y'$, intuitively, the Bayesian prior for $y \succ y'$ or $y \prec y'$ should be equal. The KL regularization proposed in Eq. (3.5) is suggesting that the preference model $P_{\theta}(y \succ y')$ should approximate a uniform distribution. Therefore, we use the KL divergence between the preference model and the uniform distribution to regularize this distance. We will include this explanation into the paper for better readability in the updated version.

---

> ### Author Response · Authors · 2024-11-21
> **Response to Reviewer AuNi (3 / 3)**
>
> > Question 1: Equation 3.4 - My understanding is that typically SFT is performed first, and then DPO, rather than optimizing both objectives jointly, as this equation implies.
>
> **Response 4**: Yes, in our framework, we combine these two objectives for the convenience of unifying those existing iterative preference fine-tuning works.  However, this does not impact the subsequent theory analysis of the rewarding bias issue. Besides, as indicated in Algorithm 1, we first apply SFT training and then DPO training in our implementations, which aligns with the approach used in original SRLM[7].
>
> [7] Yuan, Weizhe, et al. "Self-Rewarding Language Models." Forty-first International Conference on Machine Learning.
>
>
> ---
>
> >Question 2: grammar mistakes & other nits
>
> **Response 5**: Thank you for pointing out those grammar mistakes and helpful suggestions on polishing this paper. We will integrate all of them into the revised paper. We believe this revision will enhance both the readability and clarity of the paper. Additionally, we want to clairfy that $[k] = \{1, 2, \dots, k\}$ represents for the set of positive integer values, while sets denoted as calligraphic uppercase letters (e.g., $\mathcal{D}$) refer to a general sets whose elements and size are context-dependent. We have revised the descriptions to avoid potential confusion.

---

> ### Comment · Reviewer_AuNi · 2024-11-23
>
> > This indicates that if we use the same, single model for reward distillation, the model can hardly be improved.
>
> Thanks, I acknowledge that since your rewards are coming from the intrinsic reward model of the policy, this is different from the setting of prior work, and much of it does not apply directly. This also appears to be different from the original SRLM paper referenced as a baseline, which proposes using a special prompt for generating rewards ("LLM-as-a-Judge").
>
> Thanks for the discussion, and I apologize for my confusion. I now have a revised opinion of the paper:
>
> - Within the proposed scope of using the intrinsic reward model of the policy to generate rewards for self-improvement, the methods appear to be novel, as I am not aware of prior work that studies this setting. From a presentation perspective, this difference from prior work does not seem to be emphasized as significantly as it should be. It doesn't appear to be mentioned until Line 167 in section 3.1. If this is indeed the first paper proposing to use intrinsic rewards from the policy for self-improvement (and if this is why methods from prior work are not applicable), that seems significant and worth clarifying for the reader earlier and more prominently.
>
> - The idea of using the intrinsic reward model from the policy as a source of rewards for improving the policy does not seem intuitively sound, but the authors demonstrate that their method works empirically, at least to the extent that it can nearly match using a reasonably strong external RM as a source of rewards. (Calling this baseline "oracle" seems misleading)
>
> - If the method were instead using a source of rewards different from the policy (i.e. the SRLM setting using "LLM-as-a-judge", or a different RM), then there are many methods from prior work that are relevant and could potentially serve as effective regularization.
>
> It's not clear to me how interesting the proposed scope is to the community (why not just get preference judgements from a stronger RM), but I am raising my score because some of the weaknesses in my original review have been resolved.

---

> > ### Author Response · Authors · 2024-11-24
> > **Additional Response to Reviewer AuNi**
> >
> > Thank you very much for your time and efforts reviewing our response and engaging in the discussion! We greatly appreciate your recognition of the novelty of CREAM within the used scope, empirical effectiveness of CREAM and the decision to raise the score. We are happy to further clarify the SRLM settings and rewarding sources, and how the designs of CREAM fit well in such case.
> >
> > > Self-rewarding Language Model (SRLM) setting & why not generating preference judgment (annotations) from external stronger RMs?
> >
> > The SRLM setting aims at exploring a self-improvement method for aligning LLMs. We think it is meaningful to the community since it explores an approach to continuously improve the LLM by itself, **especially for the application scenarios that external stronger RMs are not available**. There are some realistic cases can be considered to apply SRLM:
> >
> > * Low-resource domain that have no existing available reward models or existing general reward models have distribution-shift problem. For example, GPT-4 may not handle well on extremely low-resource languages. In such case, SRLM can be a good choice for self-improvements.
> > * Privacy senstive and edge computing scenarios that the deployed models often possess user's personal privacy data or need to compute offline, thus these models cannot utilize external reward models for continuous improvements. And training an internal reward model can also be hard due to lack of annotations.
> >
> > Due to the limited computational resources, though our experiments are mainly conducted on 7B level LLMs (we do provide some results with 13B LLMs in Table 11 of the updated paper), we believe that CREAM is beneficial to the community and realistic applications where external reward models are not available. Additionally, our findings and method are **theoretically applicable to LLMs with any size**.
> >
> >
> > > why using intrinsic (DPO) rewards instead of LLM-as-a-Judge (prompt rewarding)
> >
> > We want to highlight our findings that prompt rewarding (LLM-as-a-Judge) may not be effective for 7B level LLMs since they have relatively weaker capability to possess complex judgment ability. As shown in the Figure 4 of the updated paper, the rewarding accuracy of prompt rewarding (P-SRLM) in both RewardBench and in-domain curated preference data remain lower than the adopted DPO rewarding (intrinsic rewards). Additionally, the inferior performance of P-SRLM on downstream tasks in Figure 5 of the updated paper also indicate the inapplicability of LLM-as-a-Judge. That means, using LLM-as-a-Judge to serve as a different source of rewards to achieve the regularization may not be successful.
> >
> > To further compare the effectiveness of applying LLM-as-a-Judge, we provide results of CREAM and distilled DPO using LLM-as-a-Judge (prompt-rewarding, denoted as P-) below. We can find that (1) prompt-rewarding is not effective as DPO rewarding; (2) The degradation of Distilled DPO in M3 iteration indicates its inapplicability for continous improvements.
> >
> > | Method        | Iteration | Arc-Easy | Arc-Challenge | OpenBookQA | SIQA  | GSM8K | Average |
> > | ------------- | --------- | -------- | ------------- | ---------- | ----- | ----- | ------- |
> > | SRLM          | M2        | 87.79    | 80.38         | 87.80      | 70.95 | 78.01 | 80.99   |
> > | P-SRLM        | M2        | 84.64    | 76.79         | 80.40      | 67.81 | 78.47 | 77.62   |
> > | Distilled DPO | M2        | 87.20    | 80.12         | 85.40      | 69.65 | 79.76 | 80.43   |
> > | P-CREAM       | M2        | 87.67    | 78.58         | 86.40      | 68.47 | 79.76 | 80.18   |
> > | CREAM         | M2        | 88.89    | 80.89         | 88.00      | 69.79 | 81.04 | 81.72   |
> > | SRLM          | M3        | 87.17    | 81.23         | 87.30      | 70.37 | 77.48 | 80.71   |
> > | P-SRLM        | M3        | 83.75    | 76.28         | 80.20      | 66.63 | 78.99 | 77.17   |
> > | Distilled DPO | M3        | 86.15    | 79.01         | 85.00      | 68.68 | 78.62 | 79.49   |
> > | P-CREAM       | M3        | 86.49    | 78.33         | 87.40      | 69.70 | 81.05 | 80.59   |
> > | CREAM         | M3        | 89.52    | 83.36         | 90.20      | 72.06 | 81.73 | 83.37   |
> >
> > > clarification about "Oracle" variant
> >
> > The "Oracle" variant uses a strong external reward model as the reward source (provider). As illustrated in Sec. 4.1 Experimental Setup (L374-L375), **we also use the ground truth labels of the downstream task to assist it in the rewarding in this "Oracle" variant** (e.g., provide the correct answer in the judgment for reference), in order to make the rewards as accurate as possible. That is the reason why we prefer name it as the "Oracle".

---

> > > ### Comment · Reviewer_AuNi · 2024-11-25
> > >
> > > Thanks again for your very thorough responses (to me, as well as to the other reviewers). I hope that the review and discussion process has been useful for improving the presentation of the paper and for helping to clarify various aspects of the methods through the additional ablations and comparisons.
> > >
> > > While I still have some concerns about the motivation for the specific problem setting being studied, my main concerns have been addressed and my score has been updated.

---

> > > > ### Author Response · Authors · 2024-11-26
> > > > **Thank you for your recognition!**
> > > >
> > > > Thank you very much for your engagement and valuable feedback throughout the discussion. We really appreciate your efforts and time dedicated to reviewing our paper. Your insightful comments have greatly improved the quality and presentation of this paper.
> > > >
> > > > We are happy to see our responses and additional experimental results have addressed most of your concerns. Thank you once again for your time, efforts, and the decision to raise the score!

---

### Official Review · Reviewer_Nh1y · 2024-11-03

**Soundness:** 3
**Presentation:** 3
**Contribution:** 2
**Rating:** 6
**Confidence:** 4

**Summary:**

This paper attempts to propose a more consistent reward learning strategy within self-rewarding LLM settings where the same model acts as the policy generating sample responses as well as a reward model that ranks those responses. Their framework called  Consistency Regularized sElf-Rewarding lAnguage Models (CREAM) leverage reward consistency signals to make their models learn from more reliable preference
data. Their experiments suggests that CREAM-trained models outperform SRLMs as well as external reward models especially with smaller models such as LLaMa2 and 3.

**Strengths:**

1. The methodology section is well described with mathematical analyses for readers.
2. Formulating the challenges and corresponding analysis is organized and explained well.
3. Various experiments are conducted to show the performance of CREAM with models like LLama2 and 3, including comparison with reasonable baselines like SRLMs and external reward models.

**Weaknesses:**

1. In the line #244, there is a lack of information about the reason for using reversed
preference order.
2. Once again, in the line #328, The reasons for preparing a reverse DPO dataset and swapping the
best response with the worst response are somewhat unclear.

**Questions:**

Questions:
1. In the line #445 - 447, It describes as “SRLM with prompt rewarding is not effective”, but
the performances are increasing on the GSM8K dataset in the Table 3. Can you explain
why this could be the case?
2. In Table 4, the performance of CREAM dis not improve when we compared to Llama2
M1 on the Arc-E, Arc-C, OBQA, and SIQA datasets. What are your thoughts on why it
didn’t improve, even though CREAM showed better performance compared to Llama3
M1?

3. Regarding the reversal of the DPO dataset, if \( C = 0 \) indicates an inverse correlation between $\theta_{t-1}$ and $\theta_{t}$, isn’t learning from this strategy placing too much confidence in the assumption that reverse preferences are meaningful? This concern arises particularly when the initial inconsistency between $\theta_{t-1}$ and $\theta_{t}$ might be due to stochasticity rather than an actual flaw in the self-rewarding model being trained. As such, did the authors conduct experiments to analyze how the trend of inconsistency evolves between consecutive iterations, especially at later stages of training (e.g., around $t = 5 $ or  $t = 6$), to see if this inconsistency stabilizes later in training?



presentation/typos:

Abstract: “there is no guarantee on the accurate of the rewarding and ranking” —> accuracy
Line 082: “we first formulates a generalized iterative preference” —> formulate
Line 139: “The objective is to iteratively minimizing the following loss” —> minimize
Line 522: “emphasizing reliable preference data and avoiding overconfident
in preference labeling.” —> overconfidence

---

> ### Author Response · Authors · 2024-11-21
> **Response to Reviewer Nh1y (1 / 2)**
>
> Thank you for your valuable feedback to help us improve this paper! We would like to answer your question by point-to-point response. We greatly appreciate it if you could let us know if our responses address your concerns.
>
> > **Weakness 1**: In the line #244, there is a lack of information about the reason for using reversed preference order.
>
> **Response 1**: Since our training is based on the preference data, it is intuitive to adopt a reversed preference order if there is evidence that the annotated preference data is reversed.  In our implementation, we actually apply the label smoothing to the DPO training using $C * DPO + (1-C) * RDPO$, where $C$ acts as the label smoothing factor. Here, Label smoothing is a well-known technique to help regularize the model training, enhancing the generalization and stabilizing the training process[1,2,3]. We will revise this paragraph to make the explanation more intuitive in the updated version.
>
> [1] Müller, Rafael, Simon Kornblith, and Geoffrey E. Hinton. "When does label smoothing help?." Advances in neural information processing systems 32 (2019).
>
> [2] Zhang, Chang-Bin, et al. "Delving deep into label smoothing." IEEE Transactions on Image Processing 30 (2021): 5984-5996.
>
> [3] Lukasik, Michal, et al. "Does label smoothing mitigate label noise?." International Conference on Machine Learning. PMLR, 2020.
>
>
> ---
>
> > **Weakness 2**: Once again, in the line #328, The reasons for preparing a reverse DPO dataset and swapping the best response with the worst response are somewhat unclear.
>
> **Response 2**: As explained in the above Response 1, we use the label smoothing for regularization. The RDPO requires to swap the selected and the rejected response. However, it is important to note that we do not need to prepare a separate reverse DPO dataset. Instead, we simply multiply $-1$ when calculating the DPO reward gap during training, which adds no additional computation overhead.

---

> ### Author Response · Authors · 2024-11-21
> **Response to Reviewer Nh1y (2 / 2)**
>
> > **Question 1**: In the line #445 - 447, It describes as “SRLM with prompt rewarding is not effective”, but the performances are increasing on the GSM8K dataset in the Table 3. Can you explain why this could be the case?
>
> **Response 3**:
>
> While the performance of P-SRLM (SRLM with prompt rewarding) shows a slight improvement on the GSM8K dataset during the 2nd and 3rd iterations, it significantly drops in accuracy across all other four datasets. This may be because GSM8K is a reasoning task with a clear correct/wrong criterion, making it easier for the LLM to judge and thus obtain more reliable preference data for training. However, its improvement on the GSM8K dataset (less than 1%) is very marginal compared to CREAM, and this slight gain could also be potentially attributed to randomness.
>
> ---
>
> > **Question 2**: In Table 4, the performance of CREAM dis not improve when we compared to Llama2 M1 on the Arc-E, Arc-C, OBQA, and SIQA datasets. What are your thoughts on why it didn’t improve, even though CREAM showed better performance compared to Llama3 M1?
>
> **Response 4**:
>
> As mentioned in Line 375, the Llama2 model has relatively weaker foundational performance, indicating that its alignment capability is not very strong. Since CREAM relies on DPO rewarding to annotate training data (preference pairs), its effectiveness depends heavily on the model's alignment. **Llama2's poor alignment** (even after SFT, its performance still drops on 2 datasets) **will inevitably lead to degeneration especially for the beginning iteration, i.e., M1 -> M2.** However, after the initial iteration, we find that **CREAM contributes to stabilizing its rewarding, which finally helps it improve for the next iteration, i.e., M2 -> M3.**
>
> ---
>
> > **Question 3**: Regarding the reversal of the DPO dataset, if ( C = 0 ) indicates an inverse correlation between $\theta_{t-1}$ and $\theta_{t}$, isn’t learning from this strategy placing too much confidence in the assumption that reverse preferences are meaningful? This concern arises particularly when the initial inconsistency between $\theta_{t-1}$ and $\theta_{t}$ might be due to stochasticity rather than an actual flaw in the self-rewarding model being trained. As such, did the authors conduct experiments to analyze how the trend of inconsistency evolves between consecutive iterations, especially at later stages of training (e.g., around t = 5 or t = 6), to see if this inconsistency stabilizes later in training?
>
> **Response 5**: We would like to highlight that the consistence between $\theta_{t-1}$ and $\theta_t$ can rarely be as low as $C \approx 0$. This is because $C = 0$ means for any preference pair $y, y'$, two consecutive model $\theta$ and $\theta_{t-1}$ give totally reversed prefernence. For the scaled Kendal Tau, that means if the ranking for 5 generated responses is $A > B > C > D > E$, to reach $C=0$, we need to change the ranking to $E > D > C > B > A$, which is very rare in SRLM. Since $\theta_t$ is trained using the prefernece data given by $\theta_{t-1}$, they tend to have similar behavior and thus such a circumstance making $C = 0$ can hardly happen in practice.
>
> Furthermore, in response to your question, we also analyze the distribution of the consistency and supply the results for 4th, 5th and 6th iterations as follows. The results below also show the consistency trend for CREAM. We can find that the initial consistency rate 0.39 is acceptable, which will not result in heavily relying on reverse preferences. Besides, the regularization of CREAM is shown to encourage the consistency of rankings and maintain stability in the later rounds of training.
>
>   | Iteration / Consistency Rate | 0-0.2         | 0.2-0.4       | 0.4-0.6       | 0.6-0.8       | 0.8-1.0       | Total       | Avg. Consistency Rate |
>   | ------------------------------ | --------------- | --------------- | --------------- | --------------- | --------------- | ------------- | ----------------------- |
>   | CREAM M2                     | 1329 (13.86%) | 3129 (32.62%) | 2977 (31.04%) | 1647 (17.17%) | 510 (5.32%)   | 9592 (100%) | 0.39                  |
>   | CREAM M3                     | 1 (0.01%)     | 10 (0.10%)    | 48 (0.50%)    | 448 (4.67%)   | 9085 (94.71%) | 9592 (100%) | 0.92                  |
>   | CREAM M4                     | 3 (0.03%)     | 24 (0.25%)    | 195 (2.03%)   | 1249 (13.03%) | 8118 (84.66%) | 9589 (100%) | 0.87                  |
>   | CREAM M5                     | 2 (0.02%)     | 10 (0.10%)    | 121 (1.26%)   | 888 (9.26%)   | 8571 (89.36%) | 9592 (100%) | 0.89                  |
>   | CREAM M6                     | 6 (0.06%)     | 109 (1.14%)   | 573 (5.97%)   | 2129 (22.20%) | 6775 (70.63%) | 9592 (100%) | 0.81                  |
>
>
> ---
>
> > Question 4: presentation/typos
>
> **Response 6**: Thank you for pointing out those grammar errors and typos. We have carefully revised and polished the paper accordingly.

---

> ### Author Response · Authors · 2024-11-24
> **We would like to hear back from Reviewer Nh1y**
>
> Dear reviewer Nh1y,
>
> Thank you for your time and efforts reviewing our paper and responses. As the discussion DDL is approaching, we would like to follow up to see if the response addresses your concerns or if you have any further questions. We would really appreciate the opportunity to discuss with you.
>
> Thank you again for your time and valuable reviews.
>
> Sincerely,
>
> Authors

---

> > ### Comment · Reviewer_Nh1y · 2024-11-26
> > **Thanks for the clarifications!**
> >
> > Thank you for your clarifications regarding my concerns. From reading your responses and the additional experiments, it appears that CREAM itself is consistently fetching performance gains across iterations. Additionally, I see that it is very unlikely for C to be zero but likely for it to be close to .9 pretty quickly, C= 0 is unlikely since previous rewards are retrieved from the immediately previous iterations and the computation using Tau requires the entire ranking to be reversed for C to be zero. As such, all my previous concerns have been resolved.
> >
> > However, upon closer reading of the paper and the discussion so far, I have another theoretical concern. I don't expect authors to run another experiment at this point but a theoretical explanation would be sufficient. My concern is this: as far as I can see, the proposed loss can be expressed as C x DPO loss + (1-C) x RDPO where C is the consistency coefficient. As such, you are essentially modeling the true preference probability ($p^*$) using a weighted DPO loss + your regularization term (say (1-C) x RDPO) . As such, how does the model learn for true preference probabilities ($p^*=1/2$) i.e., when $p(y_+ \succ y_- \mid x)$ $\sim$ $p(y_- \succ y_+ \mid x)$ $\sim$ 1/2 ?. For instance, if we replace this in the Bradley-Terry (BT) sample level preference equation (eq. 5 in [1] or eq. 6 in DPO paper), it would appear that the implicit reward difference should be zero in the typical DPO formulation.
> >
> > However, the DPO gradient is only scaled by the sigmoid of the implicit reward difference (which means DPO gradients would stay active as sigmoid of zero is 1/2). In the CREAM gradient analysis (sec C.3), CREAM gradient is shown to be scaled as $(C - \sigma(x\theta))$. In this case, if C > 1/2 and the implicit reward difference (reward gap in your paper) is zero due to BT sample level requirement --> sigmoid (zero reward gap) becomes 1/2, wouldn't the gradient be still active in CREAM updates? The reason I ask this is because empirically it seems C > 1/2 is very much a possibility very quickly (as shown in above additional results over multiple iterations) but more importantly, typical preference datasets (like Ultrafeedback) contain a non-trivial amount of samples where the rewards assigned (for two generations) are equal by a strong model (like GPT4). Acc to my reading of CREAM's motivations, it is primarily to **avoid** learning from these samples but the gradient analysis suggests that CREAM model is still penalized for such instances? In that case, wouldn't fixing C= 1/2 for such cases be more optimal to avoid learning in such cases? OTOH, if C =1 assuming there is no inconsistency (true preferences are pretty clear or  ($p^*=1$)), wouldn't CREAM underfit similar to how DPO underfits the reward function as shown in [1,2]?
> >
> >  I could be mistaken and once again, I do not expect additional experiments so a short explanation would suffice! J
> >
> >  [1] Fisch et al. 2024 (https://arxiv.org/abs/2405.19316)
> > [2] Azar, M. G., Guo, Z. D., Piot, B., Munos, R., Rowland, M., Valko, M., & Calandriello, D. (2024, April). A general theoretical paradigm to understand learning from human preferences. In International Conference on Artificial Intelligence and Statistics (pp. 4447-4455). PMLR.

---

> ### Author Response · Authors · 2024-11-27
> **Additional Response to Reviewer Nh1y**
>
> Thank you for your time and efforts reviewing our paper and responses. We are glad to hear that all your concerns have been addressed in our previous response. We would like to answer your question as follows:
>
> First, we would like to point out that unlike the existing preference dataset (e.g., Ultrafeedback[1]), in the SRLM setting, the preference data are generated by the model itself. Specifically, we randomly sample 5 responses for each question. After rewarding and ranking, we select the **top-1** scored response (A) and **bottom-1** scored response (B) to form the preference pair (A, B), ensuring the gap between A and B is as large as possible. That means it can hardly make A $\approx$ B. This preference data construction process inherently helps avoid cases where A is approximately equal to B.
>
> Second, we note that **CREAM is able to handle such extreme case** that $A \approx B$ by dynamically setting $C = 1/2$. This is because when the reward of the top-1 and bottom-1 responses among $N = 5$ responses are similar, then the remaining three responses, which fall in the middle of the rankings, would also tend to have rewards similar to both A and B (i.e., A $\approx$ others $\approx$ B). This situation suggests that the ranking can be highly uncertain (random). In such cases, the **expectation of Kendall Tau coefficient is 0**:
> $$\mathbb{E}(\tau) = \frac{\mathbb{E}(V-D)}{\binom{N}{2}} = \frac{2E[V] - \binom{N}{2}}{\binom{N}{2}} = \frac{2 \left( \binom{N}{2} \times 0.5 \right) - \binom{N}{2}}{\binom{N}{2}} = \frac{\binom{N}{2} - \binom{N}{2}}{\binom{N}{2}} = 0$$
> where V and D are concordant pairs and discordant pairs in rankings. That means, **our consistency rate C will naturally be 1/2**, suggesting that CREAM already adopts a more optimal strategy to learn from such similar preference pairs.
>
> Additionally, we supply experiment results of CREAM with C=1/2 as follows. Though universally setting C=1/2 is better than normal SRLM, it is inferior to CREAM's dynamically calculated C.
>
> | Method      | Iteration | Arc-Easy | Arc-Challenge | OpenBookQA | SIQA  | GSM8K | Average |
> | ----------- | --------- | -------- | ------------- | ---------- | ----- | ----- | ------- |
> | SRLM        | M2        | 87.79    | 80.38         | 87.80      | 70.95 | 78.01 | 80.99   |
> | CREAM+C=0.5 | M2        | 88.17    | 80.89         | 85.00      | 70.88 | 79.15 | 80.82   |
> | CREAM       | M2        | 88.89    | 80.89         | 88.00      | 69.79 | 81.04 | 81.72   |
> | SRLM        | M3        | 87.17    | 81.23         | 87.30      | 70.37 | 77.48 | 80.71   |
> | CREAM+C=0.5 | M3        | 87.29    | 80.80         | 85.00      | 67.66 | 80.06 | 80.16   |
> | CREAM       | M3        | 89.52    | 83.36         | 90.20      | 72.06 | 81.73 | 83.37   |
>
> In summary, we would like to clarify that:
> - The design of SRLM suggests that the $C$ is usually close to 1 especially for later iterations, as we demonstrated in Appendix C.
> - In the extreme case where $C = 0.5$, CREAM can indeed correctly calculate this $C$ and thus eliminate the gradient step on this situation.
>
> We hope our response could address your additional question. Thank you!
>
> [1] Cui, Ganqu, et al. "ULTRAFEEDBACK: Boosting Language Models with Scaled AI Feedback." Forty-first International Conference on Machine Learning. 2024.

---

### Official Review · Reviewer_1A7Q · 2024-11-03

**Soundness:** 3
**Presentation:** 3
**Contribution:** 3
**Rating:** 8
**Confidence:** 5

**Summary:**

The authors propose the Consistency Regularized self-rewarding Language Model (CREAM), which incorporates regularization to improve reward consistency in order to alleviate the risks of inaccurate rewards and biased training data over time. The basic idea of the method si to add a regularization of optimizing towards a balanced binary distribution, where the weight of the regularization is determined by the  self reward ranking correlation between different optimization steps. The author provides some theoretical proof and empirical results show that CREAM enhances both alignment performance and the reliability of preference data.

**Strengths:**

- The proposed method is interesting and there are theoretical and empirical proofs of effectiveness.
- The method uses the ranking correlation to evaluate the consistency or uncertainty in self rewarding.

**Weaknesses:**

- I think first the author should add some baselines such as the original form of the method which just uses the KL constraint toward the Bernoulli distribution. It would be good to break down where the improvement comes from and how much the ranking correlation helps the methods.
 - Then another very important experiment I think is that previous self-rewarding methods can not maintain the improvement beyond 3 or 4 iterations, how will your method help this? Will there still be significant improvement for the 5th or 6th iteration?
 - Another thing I am thinking is that given the final form, in the training you use a normal DPO and a reversed DPO, and weight the loss functions. Then this is basically the same as just weight the normal DPO with a weight maybe ranging from -1<->1. Why don't the author try this for example shifting the C weight and only keeping DPO loss? This would be much more efficient in implementation I assume?

**Questions:**

N/A

---

> ### Author Response · Authors · 2024-11-21
> **Response to Reviewer 1A7Q (1 / 3)**
>
> Thank you for your insightful and helpful reviews! We would like to address your concerns by point-to-point response. We would appreciate it if you could let us know whether your concerns are addressed by the responses.
>
>
> > **Weakness 1**: I think first the author should add some baselines such as the original form of the method which just uses the KL constraint toward the Bernoulli distribution. It would be good to break down where the improvement comes from and how much the ranking correlation helps the methods.
>
> **Response 1**: Thank you for your suggestion. We add two ablation models/baselines:
>   - **SRLM+KL**: SRLM with KL constraint towards the Bernoulli distribution, which introduces a simple regularization term $\lambda [\frac{\pi_\theta(y|x) }{\pi_\text{ref}(y|x)} - \frac{\pi_\theta(y'|x)}{\pi_\text{ref}(y'|x)}]^2$ to the DPO training loss.
>   - **CREAM w/o Ranking Correlation + C**: This is our method CREAM with a manually set fixed Consistency Rate ($C$). We perform a hyperparameter search to set $C$ to validate the contribution of our automatically determined $C$ via ranking correlation.
>
> The results of these two baselines are shown below. We can find that
>
> - Although SRLM+KL introduces regularization and improves performance compared to SRLM, its overall performance is weaker than CREAM. Further, CREAM maintains its advantage against SRLM+KL over multiple iterations, demonstrating that CREAM performs better than directly restricting the KL divergence.
> - Comparing CREAM w/o Ranking Correlation + C with the original CREAM, we find that the original CREAM consistently outperforms the variant with manually set $C$ values. This indicates that a fixed consistency rate is often inferior to the dynamically calculated consistency rate derived from ranking correlations.
>
>
> | Method                               | Iteration | Arc-Easy | Arc-Challenge | OpenBookQA | SIQA  | GSM8K | Average |
> | ------------------------------------ | --------- | -------- | ------------- | ---------- | ----- | ----- | ------- |
> | Initial                              | M0        | 86.29    | 80.37         | 86.00      | 68.58 | 78.01 | 79.85   |
> | SFT                                  | M1        | 86.78    | 80.14         | 86.40      | 69.50 | 78.39 | 80.24   |
> | SRLM+KL                              | M2        | 87.92    | 79.78         | 86.60      | 71.49 | 79.38 | 81.03   |
> | CREAM w/o Ranking Correlation +C=0.1 | M2        | 83.84    | 72.78         | 78.20      | 65.05 | 75.51 | 75.08   |
> | CREAM w/o Ranking Correlation +C=0.3 | M2        | 88.26    | 79.86         | 86.80      | 69.55 | 79.98 | 80.89   |
> | CREAM w/o Ranking Correlation +C=0.5 | M2        | 88.17    | 80.89         | 85.00      | 70.88 | 79.15 | 80.82   |
> | CREAM w/o Ranking Correlation +C=0.7 | M2        | 88.26    | 79.61         | 84.40      | 71.39 | 79.08 | 80.55   |
> | CREAM w/o Ranking Correlation +C=0.9 | M2        | 87.80    | 79.61         | 86.40      | 70.47 | 80.14 | 80.88   |
> | CREAM                                | M2        | 88.89    | 80.89         | 88.00      | 69.79 | 81.04 | 81.72   |
> | SRLM+KL                              | M3        | 88.38    | 80.97         | 88.20      | 71.19 | 80.29 | 81.81   |
> | CREAM w/o Ranking Correlation +C=0.1 | M3        | 89.06    | 80.46         | 85.60      | 70.88 | 79.23 | 81.05   |
> | CREAM w/o Ranking Correlation +C=0.3 | M3        | 88.09    | 80.55         | 87.20      | 71.39 | 79.23 | 81.29   |
> | CREAM w/o Ranking Correlation +C=0.5 | M3        | 87.29    | 80.80         | 85.00      | 67.66 | 80.06 | 80.16   |
> | CREAM w/o Ranking Correlation +C=0.7 | M3        | 85.65    | 77.39         | 86.00      | 68.27 | 77.94 | 79.05   |
> | CREAM w/o Ranking Correlation +C=0.9 | M3        | 84.39    | 75.17         | 83.20      | 66.02 | 78.77 | 77.51   |
> | CREAM                                | M3        | 89.52    | 83.36         | 90.20      | 72.06 | 81.73 | 83.37   |

---

> ### Author Response · Authors · 2024-11-21
> **Response to Reviewer 1A7Q (2 / 3)**
>
> > **Weakness 2**: Then another very important experiment I think is that previous self-rewarding methods can not maintain the improvement beyond 3 or 4 iterations, how will your method help this? Will there still be significant improvement for the 5th or 6th iteration?
>
> **Response 2**: To explore continuous improvements, we conducted additional experiments with SRLM (baseline method) and CREAM (ours) using Llama-3 for the 4th, 5th, and 6th iterations.  The results are reported as follows. According to the results, we have the following findings:
> * CREAM converges at the 4th iteration (M4) while the baseline method SRLM started performance degradation much earlier at the 2nd iteration (M2). This shows that CREAM helps stabilize the self-improvement training by mitigating the rewarding bias issue.
> * CREAM does not seriously harm the performance after convergence (i.e., during M5 and M6), while SRLM drastically drops the performance. This suggests that adding this consistency-based regularization is beneficial to preventing model from degeneration in the long term.
>
> **Table** ↑ and ↓ indicate the performance improvement and degradation compared to the method’s last iteration.
> | Method  | Iteration | Arc-Easy | Arc-Challenge | OpenBookQA | SIQA    | GSM8K   | Average |
> | ------- | --------- | -------- | ------------- | ---------- | ------- | ------- | ------- |
> | Initial | M0        | 86.29    | 80.37         | 86.00      | 68.58   | 78.01   | 79.85   |
> | SFT     | M1        | 86.78    | 80.14         | 86.40      | 69.50   | 78.39   | 80.24   |
> | SRLM    | M2        | 87.79 ↑  | 80.38 ↑       | 87.80 ↑    | 70.95 ↑ | 78.01 ↓ | 80.99 ↑ |
> | SRLM    | M3        | 87.17 ↓  | 81.23 ↑       | 87.30 ↓    | 70.37 ↓ | 77.48 ↓ | 80.71 ↓ |
> | SRLM    | M4        | 86.07 ↓  | 78.33 ↓       | 87.80 ↑    | 68.58 ↓ | 75.83 ↓ | 79.32 ↓ |
> | SRLM    | M5        | 84.34 ↓  | 76.53 ↓       | 85.80 ↓    | 66.84 ↓ | 64.22 ↓ | 75.55 ↓ |
> | SRLM    | M6        | 76.22 ↓  | 72.36 ↓       | 76.00 ↓    | 59.06 ↓ | 59.29 ↓ | 68.59 ↓ |
> | CREAM   | M2        | 88.89 ↑  | 80.89 ↑       | 88.00 ↑    | 69.79 ↑ | 81.04 ↑ | 81.72 ↑ |
> | CREAM   | M3        | 89.52 ↑  | 83.36 ↑       | 90.20 ↑    | 72.06 ↑ | 81.73 ↑ | 83.37 ↑ |
> | CREAM   | M4        | 89.56 ↑  | 82.68 ↓       | 90.80 ↑    | 72.93 ↑ | 82.26 ↑ | 83.65 ↑ |
> | CREAM   | M5        | 89.35 ↓  | 82.08 ↓       | 90.20 ↓    | 72.06 ↓ | 81.73 ↓ | 83.08 ↓ |
> | CREAM   | M6        | 88.85 ↓  | 81.57 ↓       | 89.60 ↓    | 71.14 ↓ | 82.49 ↑ | 82.73 ↓ |

---

> ### Author Response · Authors · 2024-11-21
> **Response to Reviewer 1A7Q (3 / 3)**
>
> > **Weakness 3**: Another thing I am thinking is that given the final form, in the training you use a normal DPO and a reversed DPO, and weight the loss functions. Then this is basically the same as just weight the normal DPO with a weight maybe ranging from -1<->1. Why don't the author try this for example shifting the C weight and only keeping DPO loss? This would be much more efficient in implementation I assume?
>
> **Response 3**: We would like to emphasize that the final form of CREAM, which combines a normal DPO and a reversed DPO, cannot simply be replaced by reducing the weight in a normal DPO. Specifically, the CREAM loss is $C \log \sigma(r(y_+) - r(y_-)) + (1 - C) \log \sigma(r(y_-) - r(y_+))$ differs fundamentally from a weighted DPO loss $C \log \sigma(r(y_+) - r(y_-))$. From the optimization perceptive, the latter weighted DPO loss behaves similar with a regular DPO loss with a smaller learning rate. In addition, since the preference probability $P(y_+ \prec y_-) = \sigma(r(y_+) - r(y_-))$ and $P(y_+ \succ y_-) = \sigma(r(y_-) - r(y_+))$, we would highlight that the CREAM loss can be kind of viewed as a cross-entropy loss $C\log P(y_+ \succ y_-) + (1 - C) \log P(y_- \succ y_+)$ with label-smoothing factor $C$. Thus, The weighted DPO loss cannot deliver such a observation.
>
> The detailed derivation is as follows:
>
> * First, the proposed loss can be expressed as $C * D + (1-C) * RD$, where $C$ is the consistency rate, $D$ and $RD$ are DPO loss and ReverseDPO loss, respectively.
> * For the DPO loss, $D = \log \sigma (a -b)$, where the sigmoid function $\sigma$ converts the reward gap $(a-b)$ into a probability $P$.  For $RD$, the reward gap is $(b - a)$. Recall that the sigmoid function has a property that $\sigma (x) = 1 - \sigma(-x)$. Using this property, we can convert the sigmoid(gap part) of $RD$ to $(1 - P)$. Thus, the $RD$ loss is $\log (1-P)$ and $D$ loss is $\log P$.
> * Combining these, we can get $C * D + (1-C) * RD = C * \log P + (1-C) * \log (1-P)$, which corresponds to a cross entropy loss for a binary classification task (binary preference judgment).
> * The proposed loss acts as a label smoothing to regularize the training. This regularization enhances the model's generalization ability and performance.
>
>
> We also take additional experiments to validate the SRLM using $\text{weight} * \text{DPO}$, where $\text{weight} \in [-1.0, 1.0]$. For SRLM + Weight method, M3 is trained based on the best checkpoint of M2 across different weights. From the results below, "**NA**" means the negative of the DPO loss leads to catastrophic forgetting, where the LLM fails to generate fluent sentences. According to the results, we observe that CREAM outperforms the SRLM+weight method, indicating its effectiveness.
>
> | Method                | Iteration | Arc-Easy | Arc-Challenge | OpenBookQA | SIQA  | GSM8K | Average |
> | --------------------- | --------- | -------- | ------------- | ---------- | ----- | ----- | ------- |
> | SRLM + Weight = -1.0  | M2        | NA       | NA            | NA         | NA    | NA    | NA      |
> | SRLM + Weight = -0.5  | M2        | NA       | NA            | NA         | NA    | NA    | NA      |
> | SRLM + Weight = -0.25 | M2        | NA       | NA            | NA         | NA    | NA    | NA      |
> | SRLM + Weight = 0.25  | M2        | 88.97    | 80.38         | 87         | 71.39 | 78.7  | 81.29   |
> | SRLM + Weight = 0.50  | M2        | 86.49    | 79.61         | 87.6       | 70.42 | 79    | 80.62   |
> | SRLM + Weight = 1.00  | M2        | 87.79    | 80.38         | 87.8       | 70.95 | 78.01 | 80.99   |
> | CREAM                 | M2        | 88.89    | 80.89         | 88         | 69.79 | 81.04 | 81.72   |
> | SRLM + Weight = -1.0  | M3        | 24.16    | 22.53         | 27.6       | 30.91 | 1.74  | 21.39   |
> | SRLM + Weight = -0.5  | M3        | NA       | NA            | NA         | NA    | NA    | NA      |
> | SRLM + Weight = -0.25 | M3        | NA       | NA            | NA         | NA    | NA    | NA      |
> | SRLM + Weight = 0.25  | M3        | 88.93    | 81.40         | 89.20      | 71.34 | 75.74 | 81.32   |
> | SRLM + Weight = 0.50  | M3        | 88.13    | 81.74         | 89.00      | 70.73 | 75.89 | 81.1    |
> | SRLM + Weight = 1.00  | M3        | 87.17    | 81.23         | 87.30      | 70.37 | 77.48 | 80.71   |
> | CREAM                 | M3        | 89.52    | 83.36         | 90.20      | 72.06 | 81.73 | 83.37   |

---

> ### Author Response · Authors · 2024-11-24
> **We would like to hear back from Reviewer 1A7Q**
>
> Dear reviewer 1A7Q,
>
> Thank you for your time and efforts reviewing our paper and responses. As the discussion DDL is approaching, we would like to follow up to see if the response addresses your concerns or if you have any further questions. We would really appreciate the opportunity to discuss with you.
>
> Thank you again for your time and valuable reviews!
>
> Sincerely,
>
> Authors

---

> > ### Comment · Reviewer_1A7Q · 2024-11-25
> > **Response**
> >
> > Given the rebuttal, I do recommend the author add the full table in response 2, and merge the results in response 1 to the main table.  This is a condition for me to raise the score to 8.

---

> > > ### Author Response · Authors · 2024-11-26
> > > **Thank you for your recognition!**
> > >
> > > Thank you very much for your time and efforts in reviewing our paper and responses. We believe your comments will enhance both the quality and presentation of our paper. Following your advice, we have updated the paper (PDF):
> > >
> > > - We added the full table (results of 4,5,6 iterations) in response 2 in Table 5 of the Appendix C.1.
> > > - We merged the results (of SRLM+KL and CREAM w/o Ranking Correlation) in response 1 into the main results Table 1. We also revised Sec. 4.1 Experimental Setup and Sec. 4.2 Main Results to include the introduction and discussion of these two methods.
> > >
> > > We sincerely appreciate your positve evaluations on our paper and the decision to raise the score to 8. Thank you for your support!

---

### Author Response · Authors · 2024-11-21
**Summary of Paper Revision**

We sincerely appreciate all reviewers for their insightful and constructive feedback. According to these comments, we have improved the paper (**new pdf uploaded**) and highlighted the main changes with blue text. Below, we summarize all changes:

**Readability and Presentation Style:**

1. All mentioned grammar errors have been fixed. And we carefully checked the whole paper. (Reviewer Nh1y and Reviewer AuNi)
2. Table 1 added three columns: "Iteration", "Reward" and "Average" for a clear presentation.
3. Figure 3 slightly changed its text alignment style.
4. Table 2 slightly changed its column arrangement and names.
5. Original Table 3 now has been converted into the Figure 5 for better presentation, the original results are still provided in Appendix Table 13.
6. In Sec. 4.3.2 and Table 3 (original Table 4), we slightly changed the description of the reference reward model, renaming it to baseline reward model (BRM), in order to correspond to the introduction in the Sec. Methodology.
7. Table 4 (original Table 5) added one column "Iteration" for a clear presentation.
8. Sec. Limitations and Future Work has been moved to Appendix A to provide additional space for the main body.

**Improved Description:**

1. We included the related work about reward hacking in L126 - 140. (Reviewer AuNi), and offer in-depth analysis between CREAM and ensemble methods in Appendix C.4. (Reviewer AuNi)
2. The description about the notations has been slightly changed to avoid confusion. (Reviewer AuNi)
3. We added the intuition about choosing the reversed preference order in L262 - L265, and provide more details in Appendix C.2. (Reviewer Nh1y and Reviewer AuNi)
4. The in-depth analysis of the loss used by CREAM including derivation and gradient analysis is provided in Appendix C.2. (Reviewer 1A7Q, Reviewer Nh1y, Reviewer LeV2)

**Additional Experimental Results:**

1. We added results of baseline methods using other KL regularizations and explore the contribution of ranking correlation into the Table 1. The introduction of these two methods and discussion for the results have been added into Sec. 4.1 Experimental Setup and Sec. 4.2 Main Results, respectively. (Reviewer 1A7Q)
2. We provided results of SRLM and CREAM for more iterations (4, 5, and 6) in Appendix C. 1 and Table 5. (Reviewer 1A7Q)
3. Results of weighted DPO are provided in Table 6, in order to validate the difference between DPO and our summed DPO loss. (Reviewer 1A7Q)
4. We analyzed the trend of the consistency rate and statistic the distribution in Appendix C. 3 and Table 7. (Reviewer Nh1y)
5. We added results of baseline methods using ensemble rewards estimation in Appendix C.4 and Table 8. (Reviewer AuNi)
6. We explored CREAM treating the consistency rate as a data property instead of model in Appendix C.5, and provided empirical results in Table 9. (Reviewer LeV2)
7. We discussed the applicability of CREAM in terms of model sizes and unaligned models and validated its effectiveness in Appendix C. 6, Table 10 and Table 11. (Reviewer LeV2)
8. We added additional results of using prompt-rewarding with CREAM and distilled-DPO in Appendix C. 7 and Table 12. (Reviewer AuNi)

---

### Meta-Review · Area_Chair_Leep · 2024-12-15

**Metareview:**

The paper addresses challenges in self-rewarding LLMs, where the same model functions as both the policy and the reward model. It highlights issues with ranking accuracy and reward systems, which lead to unreliable preference data. The authors propose CREAM, a method that introduces regularization to improve reward consistency and prevent overconfident preference labeling, thereby generating more reliable training data and enhancing alignment performance.

After reviewing the rebuttal, the authors responded effectively to the reviewers' concerns. The reviewers noted that the paper could benefit from additional background information and clearer explanations, which would improve its readability. I hope the authors will enhance the final version of the paper. My recommendation is acceptance.

**Additional Comments On Reviewer Discussion:**

Reviewer 1A7Q highlighted the lack of baselines, and the authors provided additional results to address this concern effectively.

Reviewer Nh1y raised questions about the paper's writing and detailed analysis, which the authors addressed thoroughly.

Most of Reviewer AuNi's comments focused on improving the clarity of the paper's writing.

Reviewer LeV2 pointed out a lack of experiments and analysis, and the authors provided corresponding results to resolve these concerns.

In summary, most of the reviewers' concerns have been adequately addressed. My recommendation is to accept the paper.

---

### Decision · Program_Chairs · 2025-01-22

Accept (Poster)